# Label Noise: Ignorance Is Bliss

**Yilun Zhu**
EECS
University of Michigan
allanzhu@umich.edu

**Jianxin Zhang**
EECS
University of Michigan
jianxinz@umich.edu

**Aditya Gangrade**
ECE
Boston University
gangrade@bu.edu

**Clayton Scott**
EECS, Statistics
University of Michigan
clayscot@umich.edu

## Abstract

We establish a new theoretical framework for learning under multi-class, instance-dependent label noise. This framework casts learning with label noise as a form of domain adaptation, in particular, domain adaptation under posterior drift. We introduce the concept of *relative signal strength* (RSS), a pointwise measure that quantifies the transferability from noisy to clean posterior. Using RSS, we establish nearly matching upper and lower bounds on the excess risk. Our theoretical findings support the simple *Noise Ignorant Empirical Risk Minimization (NI-ERM)* principle, which minimizes empirical risk while ignoring label noise. Finally, we translate this theoretical insight into practice: by using NI-ERM to fit a linear classifier on top of a self-supervised feature extractor, we achieve state-of-the-art performance on the CIFAR-N data challenge.

## 1 Introduction

The problem of classification with label noise can be stated in terms of random variables $X$, $Y$, and $\widetilde{Y}$, where $X$ is the feature vector, $Y \in \{1, \ldots, K\}$ is the true label associated to $X$, and $\widetilde{Y} \in \{1, \ldots, K\}$ is a noisy version of $Y$. The learner has access to i.i.d. realizations of $(X, \widetilde{Y})$, and the objective is to learn a classifier that optimizes the risk associated with $(X, Y)$.

In recent years, there has been a surge of interest in the challenging setting of instance (i.e., feature) dependent label noise, in which $\widetilde{Y}$ can depend on both $Y$ and $X$. While several algorithms have been developed, there remains relatively little theory regarding algorithm performance and the fundamental limits of this learning paradigm.

This work develops a theoretical framework for learning under multi-class, instance-dependent label noise. Our framework hinges on the concept of *relative signal strength*, which is a point-wise measure of "noisiness" in a label noise problem. Using relative signal strength to charachterize the difficulty of a label noise problem, we establish nearly matching upper and lower bounds for excess risk. We further identify distributional assumptions that ensure that the lower and upper bounds tend to zero as the sample size $n$ grows, implying that consistent learning is possible.

Surprisingly, *Noise Ignorant Empirical Risk Minimization (NI-ERM)* principle, which conducts empirical risk minimization as if no label noise exists, is (nearly) minimax optimal. To translate this insight into practice, we use NI-ERM to fit a linear classifier on top of a self-supervised feature extractor, achieving state-of-the-art performance on the CIFAR-N data challenge.

38th Conference on Neural Information Processing Systems (NeurIPS 2024).

## 2 Literature review

Theory and algorithms for classification with label noise are often based on different probabilistic models. Such models may be categorized according on how $\widetilde{Y}$ depends on $Y$ and $X$. The simplest model is symmetric noise, where the distribution of $\widetilde{Y}$ is independent of $Y$ and $X$ [Angluin and Laird, 1988]. In this case, the probability that $\widetilde{Y} = k$ is the same for all $k \neq Y$, regardless of $Y$ and $X$. In this setting, it is easy to show that minimizing the noisy excess risk (associated to the 0/1 loss) implies minimizing the clean excess risk, a property known as *immunity*. When immunity holds, there is no need to modify the learning algorithm on account of noisy labels. In other words, the learner may be *ignorant* of the label noise and still learn consistently.

A more general model is classification with label dependent noise, in which the distribution of $\widetilde{Y}$ depends on $Y$, but not $X$. Many practical algorithms have been developed over the years, based on principles including data re-weighting [Liu and Tao, 2015], robust training [Han et al., 2018, Liu et al., 2020, Hu et al., 2020, Foret et al., 2021, Liu et al., 2022] and data cleaning [Brodley and Friedl, 1999, Northcutt et al., 2021]. Consistent learning algorithms still exist, such as those based on loss correction [Natarajan et al., 2013, Patrini et al., 2017, Van Rooyen and Williamson, 2018, Liu and Guo, 2020, Zhang et al., 2022]. These approaches assume knowledge of the noise transition probabilities, which can be estimated under some identifiability assumptions [Scott et al., 2013, Zhang et al., 2021b].

In the most general setting, that of instance dependent label noise, the distribution of $\widetilde{Y}$ depends on both $Y$ and $X$. While algorithms are emerging [Cheng et al., 2021, Zhu et al., 2021, Wang et al., 2022, Yang et al., 2023], theory has primarily focused on the binary setting. Scott [2019] establishes immunity for a Neyman-Pearson-like performance criterion under a *posterior drift* model, discussed in more detail below. Cannings et al. [2020] establish an upper bound for excess risk under the strong assumption that the optimal classifiers for the clean and noisy distributions are the same.

Closest to our work, Im and Grigas [2023] derive excess risk upper and lower bounds, and reach a similar conclusion, that noise-ignorant ERM attains the lower bound up to a constant factor. Our results, based on the new concept of relative signal strength, provide a more refined analysis.

Additional connections between our contributions and prior work are made throughout the paper.

## 3 Problem statement

*Notation.* $\mathcal{X}$ denotes the feature space and $\mathcal{Y} = \{1, 2, \ldots, K\}$ denotes the label space, with $K \in \mathbb{N}$. The $K$-simplex is $\Delta^K := \{p \in \mathbb{R}^K : \forall i, p_i \geq 0, \sum p_i = 1\}$. A $K \times K$ matrix is *row stochastic* if all of its rows are in $\Delta^K$. Denote the $i$-th element of a vector $\boldsymbol{v}$ as $[\boldsymbol{v}]_i$, and the $(i, j)$-th element of a matrix $\boldsymbol{M}$ as $[\boldsymbol{M}]_{i,j}$.

In conventional multiclass classification, we observe training data $(X_1, Y_1), \ldots, (X_n, Y_n)$ drawn i.i.d. from a joint distribution $P_{XY}$. The marginal distribution of $X$ is denoted by $P_X$, and the *class posterior* probabilities $P_{Y|X=x}$ are captured by a $K$-simplex-valued vector $\boldsymbol{\eta} : \mathcal{X} \to \Delta^K$, where the $j$-th component of the vector is $[\boldsymbol{\eta}(x)]_j = \mathbb{P}(Y = j \mid X = x)$. A classifier $f : \mathcal{X} \to \mathcal{Y}$ maps an instance $x$ to a class $f(x) \in \mathcal{Y}$. Denote the risk of a classifier $f$ with respect to distribution $P_{XY}$ as $R(f) = \mathbb{E}_{(X,Y) \sim P_{XY}} \left[ \mathbb{1}_{\{f(X) \neq Y\}} \right]$. The Bayes optimal classifier for $P_{XY}$ is $f^*(x) \in \arg\max \boldsymbol{\eta}(x)$. The Bayes risk, which is the minimum achievable risk, is denoted as $R^* = R(f^*) = \inf_f R(f)$.

We consider the setting where, instead of the true class label $Y$, a noisy label $\widetilde{Y}$ is observed. The training data $(X_1, \widetilde{Y}_1), \ldots, (X_n, \widetilde{Y}_n)$ can be viewed as an i.i.d. sample drawn from a "noisy" distribution $P_{X\widetilde{Y}}$. We define $P_{\widetilde{Y}|X=x}, \widetilde{\boldsymbol{\eta}}, \widetilde{R}$ and $\widetilde{f}^*$ analogously to the "clean" distribution $P_{XY}$.

The goal of learning from label noise is to find a classifier that is able to minimize the "clean test error," that is, the risk $R$ defined w.r.t. $P_{XY}$, even though the learner has access to only corrupted training data $(X_i, \widetilde{Y}_i) \overset{\text{i.i.d.}}{\sim} P_{X\widetilde{Y}}$.

**Noise transition perspective.** Traditionally, label noise is modeled through the joint distribution of $(X, Y, \widetilde{Y})$. This joint distribution is governed by $P_X$, the clean class posterior $P_{Y|X}$, and a

matrix-valued function

$$\boldsymbol{E} : \mathcal{X} \to \{\boldsymbol{M} \in \mathbb{R}^{K \times K} : \boldsymbol{M} \text{ is row stochastic}\},$$

known as the *noise transition matrix*. The $(i, j)$-th entry of the matrix is defined as:

$$[\boldsymbol{E}(x)]_{i,j} = \mathbb{P}\left(\widetilde{Y} = j \mid Y = i, X = x\right).$$

This implies that the noisy and clean class posteriors are related by $\widetilde{\boldsymbol{\eta}}(x) = \boldsymbol{E}(x)^\top \boldsymbol{\eta}(x)$, where $^\top$ denotes the matrix transpose.

**Domain adaptation perspective.** Alternatively, label noise learning can be framed as a domain adaptation problem. In this view, $P_{X\widetilde{Y}}$ represents the source domain, and $P_{XY}$ represents the target domain. The relationship between the two domains is characterized by "posterior drift," meaning that while the source and target share the same $X$-marginal, the class posteriors (i.e., the distribution of labels given $X$) may differ [Scott, 2019, Cai and Wei, 2021, Maity et al., 2023, Liu et al., 2024]. Thus, a label noise problem can also be described by a triple $(P_X, \boldsymbol{\eta}, \widetilde{\boldsymbol{\eta}})$.

The two perspectives are equivalent, as discussed in Appendix A.1. In this work, we emphasize the domain adaptation perspective for Sections 4 and 5, and the noise transition perspective for Section 6.

## 4 Relative signal strength

To study label noise, we introduce the concept of *relative signal strength* (RSS). This is a pointwise measure of how much "signal" (certainty about the label) is contained in the noisy distribution relative to the clean distribution. Previous work [Cannings et al., 2020, Cai and Wei, 2021] has examined a related concept within the context of binary classification, under the restriction that clean and noisy Bayes classifiers are identical. Our definition incorporates multi-class classification and relaxes the requirement that the clean and noisy Bayes classifiers agree.

**Definition 1 (Relative Signal Strength)** *For any class probability vectors $\boldsymbol{\eta}, \widetilde{\boldsymbol{\eta}}$, define the* relative signal strength *(RSS) at $x \in \mathcal{X}$ as*

$$\mathcal{M}(x; \boldsymbol{\eta}, \widetilde{\boldsymbol{\eta}}) = \min_{j \in \mathcal{Y}} \frac{\max_i [\widetilde{\boldsymbol{\eta}}(x)]_i - [\widetilde{\boldsymbol{\eta}}(x)]_j}{\max_i [\boldsymbol{\eta}(x)]_i - [\boldsymbol{\eta}(x)]_j}, \tag{1}$$

*where $0/0 := +\infty$. Furthermore, for $\kappa \in [0, \infty)$, denote the set of points whose RSS exceeds $\kappa$ as*

$$\mathcal{A}_\kappa(\boldsymbol{\eta}, \widetilde{\boldsymbol{\eta}}) = \{x \in \mathcal{X} : \mathcal{M}(x; \boldsymbol{\eta}, \widetilde{\boldsymbol{\eta}}) > \kappa\}.$$

$\mathcal{M}(x; \boldsymbol{\eta}, \widetilde{\boldsymbol{\eta}})$ is a point-wise measure of how much "signal" the noisy posterior contains about the clean posterior. To gain some intuition, first notice that if the noisy Bayes classifier predicts a different class than the clean Bayes classifier, the RSS is 0 by taking $j = \arg\max \widetilde{\boldsymbol{\eta}}$ (assuming for simplicity that the $\arg\max$ is a singleton set). Now suppose the clean and noisy Bayes classifiers *do* make the same prediction at $x$, say $i^*$, and consider a fixed $j$. If

$$\frac{[\widetilde{\boldsymbol{\eta}}(x)]_{i^*} - [\widetilde{\boldsymbol{\eta}}(x)]_j}{[\boldsymbol{\eta}(x)]_{i^*} - [\boldsymbol{\eta}(x)]_j}$$

is small, it means that the clean Bayes classifier is relatively certain that $j$ is not the correct clean label, while the noisy Bayes classifier is less certain that $j$ is not the correct noisy label. Taking the minimum over $j$ gives the relative signal strength at $x$. As we formalize in the next section, a large RSS at $x$ ensures that a small (pointwise) *noisy* excess risk at $x$ implies a small (pointwise) *clean* excess risk at $x$. To gain more intuition, consider the following examples.

**Example 1** *When $\boldsymbol{\eta}(x) = [0\ 1\ 0]^\top$ and $\widetilde{\boldsymbol{\eta}}(x) = [0.3\ 0.6\ 0.1]^\top$,*

$$\mathcal{M}(x; \boldsymbol{\eta}, \widetilde{\boldsymbol{\eta}}) = \min_{j \in \mathcal{Y}} \frac{\max_i [\widetilde{\boldsymbol{\eta}}(x)]_i - [\widetilde{\boldsymbol{\eta}}(x)]_j}{\max_i [\boldsymbol{\eta}(x)]_i - [\boldsymbol{\eta}(x)]_j} = \frac{[\widetilde{\boldsymbol{\eta}}(x)]_2 - [\widetilde{\boldsymbol{\eta}}(x)]_1}{[\boldsymbol{\eta}(x)]_2 - [\boldsymbol{\eta}(x)]_1} = \frac{0.6 - 0.3}{1 - 0} = 0.3.$$

*Here, first of all, $\arg\max \boldsymbol{\eta} = \arg\max \widetilde{\boldsymbol{\eta}} = 2$, i.e., the clean and noisy Bayes classifier give the same prediction. What's more, $\mathcal{M}(x; \boldsymbol{\eta}, \widetilde{\boldsymbol{\eta}}) < 1$ because the clean Bayes classifier is absolutely certain about its prediction, while the noisy Bayes classifier is much less certain.*

**Example 2** *When* $\boldsymbol{\eta}(x) = [0\ 1\ 0]^\top$ *and* $\widetilde{\boldsymbol{\eta}}(x) = [0\ 0\ 1]^\top$,

$$\mathcal{M}(x; \boldsymbol{\eta}, \widetilde{\boldsymbol{\eta}}) = \min_{j \in \mathcal{Y}} \frac{\max_i[\widetilde{\boldsymbol{\eta}}(x)]_i - [\widetilde{\boldsymbol{\eta}}(x)]_j}{\max_i[\boldsymbol{\eta}(x)]_i - [\boldsymbol{\eta}(x)]_j} = \frac{[\widetilde{\boldsymbol{\eta}}(x)]_3 - [\widetilde{\boldsymbol{\eta}}(x)]_3}{[\boldsymbol{\eta}(x)]_2 - [\boldsymbol{\eta}(x)]_3} = \frac{1-1}{1-0} = 0.$$

*The zero signal strength results from* $\widetilde{\boldsymbol{\eta}}$ *and* $\boldsymbol{\eta}$ *leading to different predictions about* $\arg\max$.

**Example 3 (Comparison to KL divergence)** *When* $\boldsymbol{\eta}(x) = [0.05\ 0.7\ 0.25]^\top$, *and* $\widetilde{\boldsymbol{\eta}}^{(1)}(x) = [0.25\ 0.7\ 0.05]^\top$, $\widetilde{\boldsymbol{\eta}}^{(2)}(x) = [0.1\ 0.6\ 0.3]^\top$,

$$\frac{1}{\mathcal{D}_{\mathrm{KL}}\left(\boldsymbol{\eta} \,\|\, \widetilde{\boldsymbol{\eta}}^{(1)}\right)} < \frac{1}{\mathcal{D}_{\mathrm{KL}}\left(\boldsymbol{\eta} \,\|\, \widetilde{\boldsymbol{\eta}}^{(2)}\right)} \quad while \quad \mathcal{M}\left(x; \boldsymbol{\eta}, \widetilde{\boldsymbol{\eta}}^{(1)}\right) > \mathcal{M}\left(x; \boldsymbol{\eta}, \widetilde{\boldsymbol{\eta}}^{(2)}\right).$$

*Here,* $\widetilde{\boldsymbol{\eta}}^{(2)}$ *is "closer" to* $\boldsymbol{\eta}$ *in terms of KL divergence, but* $\widetilde{\boldsymbol{\eta}}^{(1)}$ *provides more information in terms of predicting the* $\arg\max$ *of* $\boldsymbol{\eta}$. *There is no conflict: KL divergence considers the similarity between two (whole) distributions, while the task of classification only focuses on predicting the* $\arg\max$.

*This also illustrates why our notion of RSS is better suited for the label noise problem than other general-purpose distance measures between distributions.*

A desirable learning scenario would be if $\mathcal{A}_\kappa(\boldsymbol{\eta}, \widetilde{\boldsymbol{\eta}}) = \mathcal{X}$ for some large $\kappa$, indicating that the signal strength is big across the entire space. Unfortunately, this ideal situation is generally not achievable. To gain some insight, consider the following result, proved in Appendix A.2.1.

**Proposition 1** $\mathcal{A}_0(\boldsymbol{\eta}, \widetilde{\boldsymbol{\eta}}) = \left\{ x \in \mathcal{X} : \arg\max \widetilde{\boldsymbol{\eta}}(x) \subseteq \arg\max \boldsymbol{\eta}(x) \right\}.$

If we assume that both $\arg\max$ sets are singletons, this result indicates that $\mathcal{A}_0$, the region with positive RSS, is the region where the true and noisy Bayes classifiers agree. Accordingly, $\mathcal{X} \setminus \mathcal{A}_0$, the zero signal region, is the region where the clean and noisy Bayes decision rules differ. The "region of strong signal," $\mathcal{A}_\kappa$, is a subset of $\mathcal{A}_0$. Since the clean and noisy Bayes classifiers will typically disagree for at least some $x$, $\mathcal{A}_0 \neq \mathcal{X}$ in general. We note that the strong assumption that $\mathcal{A}_0 = \mathcal{X}$ has been made in prior studies [Cannings et al., 2020, Cai and Wei, 2021]. Our notion of RSS relaxes this assumption and provides a unified view.

### 4.1 RSS in binary classification

We can express relative signal strength more explicitly in the binary setup. Let $\eta(x) := [\boldsymbol{\eta}(x)]_1 = \mathbb{P}(Y = 1 \mid X = x)$ and $\widetilde{\eta}(x) := [\widetilde{\boldsymbol{\eta}}(x)]_1 = \mathbb{P}(\widetilde{Y} = 1 \mid X = x)$. In standard binary classification, the *margin* [Tsybakov, 2004, Massart and Nédélec, 2006], defined as $\left|\eta(x) - \frac{1}{2}\right|$, serves as a pointwise measure of signal strength. Our notion of relative signal strength (RSS) can be interpreted as an extension of this concept in the context of label noise learning.

**Proposition 2** *In the binary setting, for* $\kappa \geq 0$,

$$\mathcal{M}(x; \eta, \widetilde{\eta}) = \max\left\{ \frac{\widetilde{\eta}(x) - \frac{1}{2}}{\eta(x) - \frac{1}{2}}, \ 0 \right\}, \qquad and \qquad \mathcal{A}_\kappa(\eta, \widetilde{\eta}) = \left\{ x \in \mathcal{X} : \frac{\widetilde{\eta}(x) - \frac{1}{2}}{\eta(x) - \frac{1}{2}} > \kappa \right\}.$$

In other words, RSS can be viewed as a "relative" margin.

**Example 4** *Illustration of relative signal strength in a binary classification setup (Figure 1).*

### 4.2 Posterior Drift Model Class.

Now putting definitions together, we consider the posterior drift model $\Pi$ defined over the triple $(P_X, \boldsymbol{\eta}, \widetilde{\boldsymbol{\eta}})$. Let $\epsilon \in [0, 1], \kappa \in [0, +\infty)$, and define

$$\Pi(\epsilon, \kappa) := \left\{ (P_X, \boldsymbol{\eta}, \widetilde{\boldsymbol{\eta}}) : P_X\left(\mathcal{A}_\kappa\left(\boldsymbol{\eta}, \widetilde{\boldsymbol{\eta}}\right)\right) \geq 1 - \epsilon \right\}.$$

This is a set of triples (label noise problems) such that $\mathcal{A}_\kappa$, the region with RSS at least $\kappa$, covers at least $1 - \epsilon$ of the probability mass. In the next section, we will demonstrate that the performance within $\mathcal{A}_\kappa$ can be guaranteed, whereas learning outside the region $\mathcal{A}_\kappa$ is provably challenging.

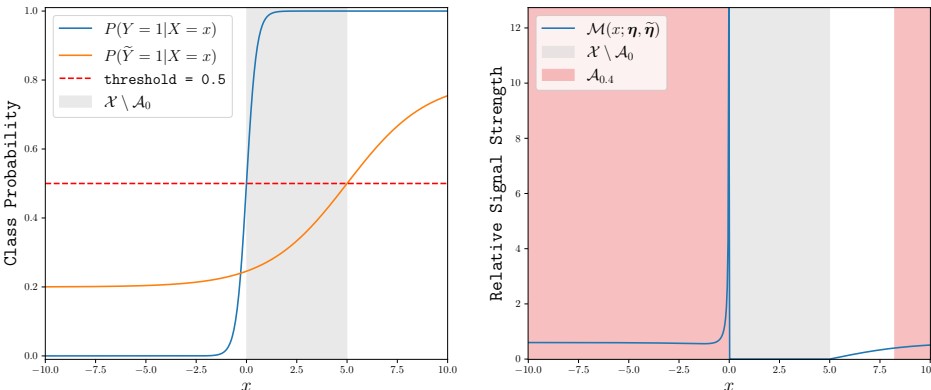

Figure 1: Illustration of relative signal strength for binary classification. *Left*: clean and noisy posteriors $[\boldsymbol{\eta}(x)]_1 = \mathbb{P}(Y = 1|X = x)$ and $[\widetilde{\boldsymbol{\eta}}(x)]_1 = \mathbb{P}(\widetilde{Y} = 1|X = x)$. *Right*: relative signal strength corresponding to these posteriors. The gray region, $x \in (0, 5)$, is where the true and noisy Bayes classifiers differ, and is also the zero signal region $\mathcal{X} \setminus \mathcal{A}_0$. The red region is $\mathcal{A}_{0.4}$, where the RSS is $> 0.4$. Note that as $x \uparrow 0$, $\mathcal{M}(x; \boldsymbol{\eta}, \widetilde{\boldsymbol{\eta}}) \uparrow \infty$, which occurs since $[\boldsymbol{\eta}(x)]_1 \uparrow 1/2$, while $[\widetilde{\boldsymbol{\eta}}]_1$ is far from $1/2$. For $x = 0^+$, the predicted labels under $\boldsymbol{\eta}$ and $\widetilde{\boldsymbol{\eta}}$ disagree, and the RSS crashes to 0.

## 5   Upper and lower bounds

In this section, we establish both upper and lower bounds for excess risk under multi-class instance-dependent label noise.

### 5.1   Minimax lower bound

Our first theorem reveals a fundamental limit: no classifier trained using noisy data can surpass the constraints imposed by relative signal strength in a minimax sense. To state the theorem, we employ the following notation and terminology. Denote the noisy training data by $Z^n = \left\{(X_i, \widetilde{Y}_i)\right\}_{i=1}^n \overset{i.i.d.}{\sim} P_{X\widetilde{Y}}$. A *learning rule* $\hat{f}$ is an algorithm that takes $Z^n$ and outputs a classifier. The risk $R(\hat{f})$ of a learning rule is a random variable, where the randomness is due to the draw $Z^n$.

**Theorem 1 (Minimax Lower Bound)** *Let $\epsilon \in [0, 1], \kappa > 0$. Then*

$$\inf_{\hat{f}} \sup_{(P_X, \boldsymbol{\eta}, \widetilde{\boldsymbol{\eta}}) \in \Pi(\epsilon, \kappa)} \mathbb{E}_{Z^n}\left[R\left(\hat{f}\right) - R(f^*)\right] \geq \frac{K-1}{K}\epsilon + \Omega\left(\frac{1}{\kappa}\sqrt{\frac{1}{n}}\right),$$

*where the* inf *is over all learning rules.*

The proof in Appendix A.2.3 offers insights into how label noise impacts the learning process: On the low RSS region ($\mathcal{X} \setminus \mathcal{A}_\kappa$), learning is difficult if not impossible, and the learner incurs an irreducible error of $(1 - 1/K)\epsilon$. On the high RSS region ($\mathcal{A}_\kappa$), the standard nonparametric rate [Devroye et al., 1996] is scaled by $1/\kappa$. These aspects determine fundamental limits that no classifier trained only on noisy data can overcome without additional assumptions.

### 5.2   Upper bound

This subsection establishes an upper bound for *Noise Ignorant Empirical Risk Minimizer (NI-ERM)*, the empirical risk minimizer trained on noisy data. This result implies that NI-ERM is (nearly) minimax optimal, a potentially surprising result given that NI-ERM is arguably the simplest approach one might consider. We begin by presenting a general result on the excess risk of any classifier, which is proved in Appendix A.2.4.

**Lemma 1 (Oracle Inequality)** *For any label noise problem* $(P_X, \boldsymbol{\eta}, \widetilde{\boldsymbol{\eta}})$ *and any classifier* $f$,

$$R(f) - R(f^*) \leq \inf_{\kappa > 0} \left\{ P_X \left( \mathcal{X} \setminus \mathcal{A}_\kappa (\boldsymbol{\eta}, \widetilde{\boldsymbol{\eta}}) \right) + \frac{1}{\kappa} \left( \widetilde{R}(f) - \widetilde{R} \left( \tilde{f}^* \right) \right) \right\}.$$

For $(P_X, \boldsymbol{\eta}, \widetilde{\boldsymbol{\eta}}) \in \Pi(\epsilon, \kappa)$, the first term is bounded by $\epsilon$. When $f$ is selected by ERM over the noisy training data, conventional learning theory implies a bound on the second term. This leads to the following upper bound for NI-ERM, whose proof is in Appendix A.2.5.

**Theorem 2 (Excess Risk Upper Bound of NI-ERM)** *Let* $\epsilon \in [0,1], \kappa > 0$. *Consider any* $(P_X, \boldsymbol{\eta}, \widetilde{\boldsymbol{\eta}}) \in \Pi(\epsilon, \kappa)$, *assume function class* $\mathcal{F}$ *has Natarajan dimension* $V$, *and the noisy Bayes classifier* $\tilde{f}^*$ *belongs to* $\mathcal{F}$. *Let* $\hat{f} \in \mathcal{F}$ *be the ERM trained on* $Z^n = \left\{ (X_i, \widetilde{Y}_i) \right\}_{i=1}^n$, *i.e.,* $\hat{f} = \arg\min_{f \in \mathcal{F}} \frac{1}{n} \sum_{i=1}^n \mathbb{1}_{\left\{ f(X_i) \neq \widetilde{Y}_i \right\}}$. *Then*

$$\mathbb{E}_{Z^n} \left[ R\left( \hat{f} \right) - R(f^*) \right] \leq \epsilon + \tilde{\mathcal{O}} \left( \frac{1}{\kappa} \sqrt{\frac{V}{n}} \right).$$

$\tilde{\mathcal{O}}$ denotes big-$\mathcal{O}$ notation ignoring logarithmic factors. The Natarajan dimension is a multiclass analogue of the VC dimension. The upper bound in Theorem 2 aligns with the minimax lower bound (Theorem 1) in both terms. For the irreducible error $\epsilon$, there is a small gap of $1/K$. This gap arises because, in the lower bound construction, the low signal region $\mathcal{X} \setminus \mathcal{A}_\kappa$ is known to the learner, whereas knowledge of $\mathcal{X} \setminus \mathcal{A}_\kappa$ is not provided to NI-ERM. If $\mathcal{A}_\kappa$ were known to the learner (an unrealistic assumption), then a mixed strategy that preforms NI-ERM on $\mathcal{A}_\kappa$ and randomly guesses on $\mathcal{X} \setminus \mathcal{A}_\kappa$ would have an upper bound with first term of $(1 - 1/K)\epsilon$, exactly matching the lower bound. Regarding the second term, there is a universal constant and a logarithmic factor between the lower and upper bounds, which is a standard outcome in learning theory.

This result is surprising as it indicates that the simplest possible approach, which ignores the presence of noise, is nearly optimal. No learning rule could perform significantly better in this minimax sense.

### 5.3 A smooth margin-condition on the relative signal strength

The previous sections have analyzed learning with label noise over the class $\Pi(\epsilon, \kappa) = \{(P_X, \boldsymbol{\eta}, \tilde{\boldsymbol{\eta}}) : P_X(\mathcal{A}_\kappa) \geq 1 - \epsilon\}$. Now, the set $\mathcal{X} \setminus \mathcal{A}_\kappa$ equals $(\mathcal{X} \setminus \mathcal{A}_0) \cup (\mathcal{A}_0 \setminus \mathcal{A}_\kappa)$. The first part of this decomposition is the region where the Bayes classifiers under the noisy and clean distributions differ, while the second is a region where these match, but the RSS is small. Naturally, while $P_X(\mathcal{X} \setminus \mathcal{A}_0)$ must be incurred as an irreducible error, one may question why the class $\Pi$ also limits the mass of $\mathcal{A}_0 \setminus \mathcal{A}_\kappa$. After all, with enough data, the optimal prediction in this region can be learned.

This issue would be resolved if there existed a $\kappa_0 > 0$ such that $P_X(\mathcal{A}_0) = P_X(\mathcal{A}_{\kappa_0})$, i.e., if $P_X(\mathcal{A}_0 \setminus \mathcal{A}_{\kappa_0}) = 0$. In fact, our lower bound from Theorem 1 uses precisely such a construction. An interesting point of comparison to this condition lies in Massart's hard-margin condition from standard supervised learning theory, which, for binary problems, demands that $P_X(|[\boldsymbol{\eta}(x)]_1 - [\boldsymbol{\eta}(x)]_2| < h) = 0$ for some $h > 0$, under which one obtains minimax excess risk bounds of $O(V/nh)$ [Massart and Nédélec, 2006]. With this lens, we can view the condition $P_X(\mathcal{A}_0 \setminus \mathcal{A}_{\kappa_0}) = 0$ as a type of *hard-margin condition on the relative signal strength* $\mathcal{M}$. This naturally motivates a smoothened version of this condition, inspired by Tsybakov's soft-margin condition [Tsybakov, 2004].

**Definition 2** *A triple* $(P_X, \boldsymbol{\eta}, \widetilde{\boldsymbol{\eta}})$ *satisfies an* $(\epsilon, \alpha, C_\alpha)$-*smooth relative signal margin condition with* $\epsilon \in [0,1], \alpha > 0, C_\alpha > 0$ *if*

$$\forall \kappa > 0, \ P_X(\mathcal{M}(x; \boldsymbol{\eta}, \widetilde{\boldsymbol{\eta}}) \leq \kappa) \leq C_\alpha \kappa^\alpha + \epsilon.$$

*Further, we define* $\Pi'(\epsilon, \alpha, C_\alpha)$ *as the set of triples* $(P_X, \boldsymbol{\eta}, \widetilde{\boldsymbol{\eta}})$ *that satisfy an* $(\epsilon, \alpha, C_\alpha)$-*smooth relative signal margin condition.*

We show in Appendix A.2.7 that the techniques of Section 5.2 yield the following result for $\Pi'$.

**Theorem 3** *Let $\epsilon \in [0,1], \alpha > 0, C_\alpha > 0$. Consider any $(P_X, \boldsymbol{\eta}, \widetilde{\boldsymbol{\eta}}) \in \Pi'(\epsilon, \alpha, C_\alpha)$, assume function class $\mathcal{F}$ has Natarajan dimension $V$, and the noisy Bayes classifier $\tilde{f}^*$ belongs to $\mathcal{F}$. Let $\hat{f} \in \mathcal{F}$ be the ERM trained on $Z^n = \left\{(X_i, \widetilde{Y}_i)\right\}_{i=1}^n$. Then*

$$\mathbb{E}_{Z^n}\left[R\left(\hat{f}\right) - R\left(f^*\right)\right] \leq \epsilon + \inf_{\kappa > 0}\left\{C_\alpha \kappa^\alpha + \tilde{\mathcal{O}}\left(\frac{1}{\kappa}\sqrt{\frac{V}{n}}\right)\right\} = \epsilon + \tilde{\mathcal{O}}\left(n^{-\alpha/(2+2\alpha)}\right).$$

Compared to Theorem 2, we see that the rate of the second term is slightly slower, which is consistent with standard learning theory where Massart's hard margin assumption leads to faster rates than Tsybakov's. The advantage of the smooth relative margin is that the irreducible term in the above theorem is exactly $P_X(\mathcal{X} \setminus \mathcal{A}_0)$, which has a clearer meaning as it measures the mismatch between clean and noisy Bayes classifiers. Further, notice that the NI-ERM algorithm does not need information about $\alpha$, and thus the result is adaptive to both $\alpha$, and to the optimal $\kappa$ for each value of $\alpha$, as a consequence of the oracle inequality of Lemma 1.

More broadly, Theorem 3 illustrates the flexibility of our conceptualization of label noise problems through RSS. The RSS $\mathcal{M}$ characterizes the irreducible error in label noise learning, similar to how the regression function $\boldsymbol{\eta}$ characterizes excess risk in standard learning. Thus, standard theoretical frameworks can be adapted to the noisy label problem via the *relative signal*.

## 6 Conditions that ensure noise immunity

The minimax lower bound in the previous section revealed a negative outcome, indicating that no method can do well in the low signal region. Nevertheless, numerous empirical successes have been observed even under significant label noise. This is not mere coincidence. In this section, we will illustrate that the high signal region $\mathcal{A}_\kappa$ can indeed cover the entire input space $\mathcal{X}$ even under massive label noise, albeit with the constraint $\mathcal{A}_\kappa \subseteq \mathcal{A}_0$ as stated in Proposition 1. This not only explains past empirical successes, but also gives a rigorous condition on the consistency of NI-ERM.

This section will delve into the study of noise transition matrix $\boldsymbol{E}$ and establish precise conditions that lead to $\mathcal{A}_0 = \mathcal{X}$. These conditions are linear algebraic conditions on $\boldsymbol{E}$ that ensure $\arg\max \widetilde{\boldsymbol{\eta}}(x) = \arg\max \boldsymbol{\eta}(x)$. As a result, we can infer that in a 10-class classification problem, even with up to $90\%$ of training labels being incorrect, the NI-ERM can still asymptotically achieve Bayes accuracy. In the upcoming definition, we introduce the concept of noise immunity, wherein the optimal classifiers remain unaffected by label noise [Menon et al., 2015, Scott, 2019].

**Definition 3 (Immunity)** *We say that a $K$-class classification problem $(P_X, \boldsymbol{\eta}, \widetilde{\boldsymbol{\eta}})$ is immune to label noise if $\forall x \in \mathcal{X}, \arg\max \widetilde{\boldsymbol{\eta}}(x) = \arg\max \boldsymbol{\eta}(x)$.*

Notice that due to Proposition 1, if a problem is immune, then $\mathcal{A}_0 = \mathcal{X}$. We now provide necessary and sufficient conditions on noise transition matrix $\boldsymbol{E}$ that ensure noise immunity. We begin by considering distribution $P_{XY}$ with zero Bayes risk, that is, where $\boldsymbol{\eta}$ is one-hot almost surely. A matrix is defined as diagonally dominant if, for each row, the diagonal element is the unique maximum.

**Theorem 4 (Immunity for Zero-error Distribution)** *If $P_{XY}$ has Bayes risk of zero, then immunity holds if and only if for all $x$, the noise transition matrix $\boldsymbol{E}(x)$ is diagonally dominant.*

**Remark** *For a zero-error distribution $P_{XY}$, even corrupted with instance-dependent label noise, achieving the Bayes risk is still feasible with a noise rate $\mathbb{P}\left(\widetilde{Y} \neq Y\right)$ up to $\frac{K-1}{K}$. This highlights that the task of classification itself is robust to label noise, specially when the clean $\boldsymbol{\eta}$ is well-separated.*

The above result relies on strong assumptions about the distribution $P_{XY}$. Now, we present a result that applies to any distribution, which, as a trade-off, turns out to impose more requirements on $\boldsymbol{E}$.

**Theorem 5 (Universal Immunity)** *For any choice of $P_{XY}$, immunity holds*

$$\iff \quad \exists\, e(x) > 0 \text{ s.t. } \forall x \in \mathcal{X}, \, [\boldsymbol{E}(x)]_{i,j} = \begin{cases} \frac{1}{K} + e(x) & i = j \\ \frac{1}{K} - \frac{e(x)}{K-1} & i \neq j. \end{cases}$$

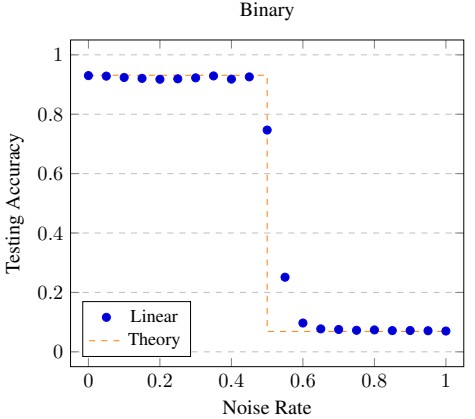

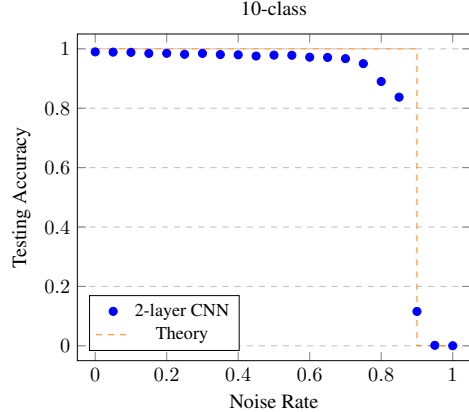

(a) Logistic regression trained on 2D gaussian mixture data with different levels of symmetric noise

(b) 2-layer CNN trained on MNIST with different levels of symmetric noise

Figure 2: Data simulation that verifies noise immunity. For binary, the turning point is at noise rate $\mathbb{P}\big(\widetilde{Y} \neq Y\big) = 0.5$. For 10-class, the turning point is at $\mathbb{P}\big(\widetilde{Y} \neq Y\big) = 0.9$.

Previous works [Ghosh and Kumar, 2017, Menon et al., 2018, Oyen et al., 2022] have established that symmetric label noise is sufficient for immunity. Our contribution advances this understanding by demonstrating that such noise conditions are not only sufficient but also necessary. Specifically, under symmetric label noise, learning towards the Bayes classifier is feasible as long as the proportion of wrong labels does not exceed $\frac{K-1}{K}$. Furthermore, this transition is abrupt: when $\mathbb{P}\big(\widetilde{Y} \neq Y\big) < \frac{K-1}{K}$, $\mathcal{A}_0 = \mathcal{X}$, but when $\mathbb{P}\big(\widetilde{Y} \neq Y\big) \geq \frac{K-1}{K}$, $\mathcal{A}_0 = \emptyset$. Consequently, we expect to see a sudden drop in performance when noise rate passes the threshold.

The rationale behind the necessity of $\boldsymbol{E}(x)$ taking this specific form is that it redistributes the probability mass of $\boldsymbol{\eta}$ in a "uniform" manner. This constraint arises because $\boldsymbol{E}(x)$ cannot favor any classes besides the true class. For instance, consider $\boldsymbol{\eta}(x) = \big[\frac{1}{K} + \delta \ \frac{1}{K} - \delta \ \frac{1}{K} \cdots \frac{1}{K}\big]^{\top}$ for some small $\delta > 0$, a "non-uniform" $\boldsymbol{E}(x)$ would alter the $\arg\max$.

The above theorems demonstrate that signal strength at $x$ can still be high even under massive label noise $\mathbb{P}\big(\widetilde{Y} \neq Y\big)$, and, in essence, it is the discrete nature of the classification problem that allows robustness to label noise. When immunity holds, the irreducible error in Theorem 2 vanishes, therefore NI-ERM becomes a consistent learning rule. We validate this through data simulations presented in Figure 2, where we systematically flip labels uniformly and observe the corresponding changes in the testing accuracy of NI-ERM. The simulation results align closely with the theoretical expectations: NI-ERM achieves near-Bayes risk performance until a certain noise threshold is reached, beyond which the testing performance sharply deteriorates.

## 7 Practical implication

The modern practice of machine learning often involves training a deep neural network. In complex tasks involving noisy labels, the naïve NI-ERM is often outperformed by state-of-the-art methods by a significant extent [Li et al., 2020, Xiao et al., 2023]. This is consistent with the finding that directly training a large neural network on noisy data frequently leads to overfitting [Zhang et al., 2021a].

Yet this is not grounds for abandoning NI-ERM altogether as a practical strategy. Instead of using NI-ERM for end-to-end training of a deep neural network, we instead propose the following simple, two-step procedure, termed 'feature extraction + NI-ERM'.

1. Perform feature extraction using any method (e.g., transfer learning or self-supervised learning) that does not require labels.

2. Learn a simple classifier (e.g., a linear classifier) on top of these extracted features, using the noisily labelled data, in a noise-ignorant way.

This approach has three advantages over full network training. First, it avoids the potentially negative impact of the noisy labels on the extracted features. Second, it enjoys the inherent robustness of fitting a simple model (step 2) on noisy data, which we observed in Figure 2. Third, it avoids the need to tune hyperparameters of the feature extractor using noisy labels. We note that a "self-supervised + simple approach" to learning was previously studied by Bansal et al. [2021], although their focus was on generalisation properties without label noise. We also acknowledge that the practical idea of ignoring label noise is not new [Ghosh and Lan, 2021], but the full power of this approach has not been previously recognized. For example, prior works often combine this approach with additional steps or employ early stopping to mitigate the effects of noise [Zheltonozhskii et al., 2022, Xue et al., 2022].

Remarkably, this two-step approach attains extremely strong performance. We conducted experiments [1] on the CIFAR image data under two scenarios: synthetic label flipping (symmetric noise) and realistic human label errors [Wei et al., 2022], as shown in Figure 3. We examine three different feature extractors: the DINOv2 foundation model [Oquab et al., 2023], ResNet-50 features extracted from training on ImageNet [He et al., 2016], and self-supervised ResNet-50 using contrastive loss [Chen et al., 2020]. We also compared to a simple linear model trained on the raw pixel intensities, and a ResNet-50 trained end-to-end. We observed that ResNet-50 exhibits degrading performance with increasing noise, consistent with previous findings [Zhang et al., 2021a, Mallinar et al., 2022]. The linear model demonstrates robustness to noise, but suffers from significant approximation error.

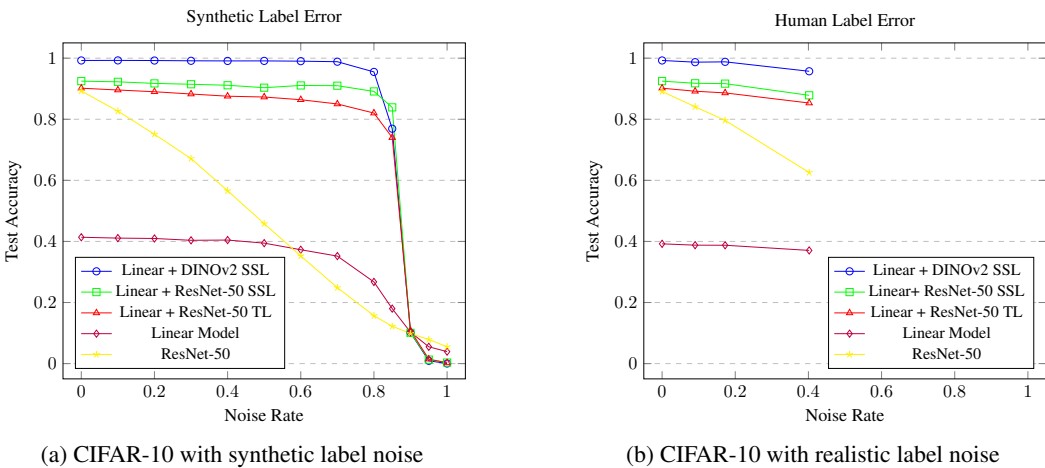

(a) CIFAR-10 with synthetic label noise
(b) CIFAR-10 with realistic label noise

Figure 3: A linear model trained on features obtained from either transfer learning (pretrained ResNet-50 on ImageNet [He et al., 2016] ), self-supervised learning (ResNet-50 trained on CIFAR-10 images with contrastive loss [Chen et al., 2020]), or a pretrained self-supervised foundation model DINOv2 [Oquab et al., 2023] significantly boosts the performance of the original linear model. In contrast, full training of a ResNet-50 leads to overfitting.

Conversely, the FE+NI-ERM approach enjoys the best of both worlds. Regardless of how the feature extraction is carried out, the resulting models exhibit robustness to label noise, while the overall accuracy depends entirely on the quality of the extracted features. This is illustrated in Figure 3, where the flatness of the accuracy curves as noise increases indicates the robustness, and the intercept at zero label noise is a measure of the feature quality. Importantly, this property holds true even under realistic label noise of CIFAR-N [Wei et al., 2022]. In fact, we find that using the DINOv2 [Oquab et al., 2023] extracted features in our FE+NI-ERM approach yields state of the art results on the CIFAR-10N and CIFAR-100N benchmarks, across the noise levels, as shown in Table 1.

We emphasize that the only hyperparameters of our model are the hyperparameters of the linear classifier, which are tuned automatically using standard cross-validation on the noisy labels. This contrasts to the implementation of many methods on the CIFAR-N leaderboard (http://noisylabels.com/)

---

[1]Code is available at: https://github.com/allan-z/label_noise_ignorance.

[2], where the hyperparameters are hard-coded. Furthermore, our approach does not rely on data augmentation. Additional experiments, detailed in Appendix A.4 , include comparisons with the 'linear probing, then fine-tuning' approach [Kumar et al., 2022], the application of different robust learning strategies on DINOv2 features, and results on synthetic instance-dependent label noise.

Overall, the strong performance, the simplicity of the approach and the lack of any untunable hyperparameters highlights the effectiveness of FE+NI-ERM, and indicates the value of further investigation into its properties.

Table 1: Performance comparison with CIFAR-N leaderboard (`http://noisylabels.com/`) in terms of testing accuracy. "Aggre", "Rand1", ..., "Noisy" denote various types of human label noise. We compare with four methods that covers the top three performance for all noise categories: ProMix [Xiao et al., 2023], ILL [Chen et al., 2023], PLS [Albert et al., 2023] and DivideMix [Li et al., 2020]. Our approach, a Noise Ignorant linear model trained on features extracted by the self-supervised foundation model DINOv2 [Oquab et al., 2023] achieves new state-of-the-art results, highlighted in bold. We employed Python's sklearn logistic regression and cross-validation functions without data augmentation; the results are deterministic and directly reproducible.

| Leaderboard | CIFAR-10N | | | | | CIFAR-100N |
|---|---|---|---|---|---|---|
| Methods | Aggre | Rand1 | Rand2 | Rand3 | Worst | Noisy |
| ProMix | $97.65 \pm 0.19$ | $97.39 \pm 0.16$ | $97.55 \pm 0.12$ | $97.52 \pm 0.09$ | $\mathbf{96.34 \pm 0.23}$ | $73.79 \pm 0.28$ |
| ILL | $96.40 \pm 0.03$ | $96.06 \pm 0.07$ | $95.98 \pm 0.12$ | $96.10 \pm 0.05$ | $93.55 \pm 0.14$ | $68.07 \pm 0.33$ |
| PLS | $96.09 \pm 0.09$ | $95.86 \pm 0.26$ | $95.96 \pm 0.16$ | $96.10 \pm 0.07$ | $93.78 \pm 0.30$ | $73.25 \pm 0.12$ |
| DivideMix | $95.01 \pm 0.71$ | $95.16 \pm 0.19$ | $95.23 \pm 0.07$ | $95.21 \pm 0.14$ | $92.56 \pm 0.42$ | $71.13 \pm 0.48$ |
| FE + NI-ERM | $\mathbf{98.69 \pm 0.00}$ | $\mathbf{98.80 \pm 0.00}$ | $\mathbf{98.65 \pm 0.00}$ | $\mathbf{98.67 \pm 0.00}$ | $95.71 \pm 0.00$ | $\mathbf{83.17 \pm 0.00}$ |

**RSS for realistic human label error.** To calculate the RSS under realistic human label error, we train two linear classifiers on DINOv2 features under clean and noisy labels and use the models' predictions as estimates for the class probabilities $\boldsymbol{\eta}$ and $\widetilde{\boldsymbol{\eta}}$. Despite the high overall noise rate in CIFAR-10N "Worst" labels, with $\mathbb{P}(Y \neq \widetilde{Y}) = 40.21\%$, we conjecture that the region where there is no signal, $\mathcal{X} \setminus \mathcal{A}_0$, covers only a small portion of the probability mass ($\epsilon \leq 4\%$). Furthermore, the cumulative distribution of the estimated RSS can be upper-bounded by a polynomial $C_\alpha \kappa^\alpha + \epsilon$, supporting the validity of the smooth relative signal margin condition introduced in Section 5.3.

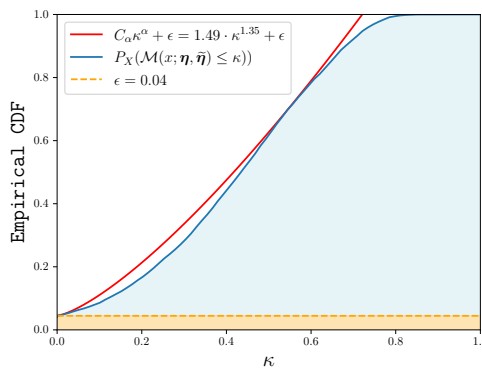

Figure 4: Empirical CDF of estimated RSS for CIFAR-10N, evaluated on test data.

## 8 Conclusions

This work presents a rigorous theory for learning under multi-class, instance-dependent label noise. We establish nearly matching upper and lower bounds for excess risk and identify precise conditions for classifier consistency. Our theory reveals the (nearly) minimax optimality of Noise Ignorant Empirical Risk Minimizer (NI-ERM). To make this theory practical, we provide a simple modification leveraging a feature extractor with NI-ERM, demonstrating significant performance enhancements. A limitation of this work is that our methodology warrants more extensive experimental evaluation.

---

## Acknowledgements

This work was supported in part by the National Science Foundation under award 2008074 and the Department of Defense, Defense Threat Reduction Agency under award HDTRA1-20-2-0002. The authors thank Zixuan Huang, Yihao Xue for helpful discussions and Raj Rao Nadakuditi for feedback during a course project in which some early experiments in this paper were conducted. We also thank the anonymous reviewers for their suggestions, especially the reviewer who provided Example 3.

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

# A    Appendix / supplemental material

## A.1    Equivalence of noise transition and domain adaptation perspectives

The noise transition perspective models the joint distribution of $(X, Y, \widetilde{Y})$, which can be characterized as:

$$P_{X,Y,\widetilde{Y}} = P_X \underbrace{P_{Y|X}}_{\boldsymbol{\eta}} \underbrace{P_{\widetilde{Y}|Y,X}}_{\boldsymbol{E}}$$

Thus, by specifying $P_X$, $\boldsymbol{\eta}$, and $\boldsymbol{E}$, we obtain a triple $(P_X, \boldsymbol{\eta}, \widetilde{\boldsymbol{\eta}})$ with $\widetilde{\boldsymbol{\eta}}(x) = \boldsymbol{E}(x)^\top \boldsymbol{\eta}(x)$.

In contrast, the domain adaptation perspective views label noise problems directly as a triple $(P_X, \boldsymbol{\eta}, \widetilde{\boldsymbol{\eta}})$, bypassing the explicit modeling of the noise transition matrix $\boldsymbol{E}$.

If no assumptions are made about the form of $\boldsymbol{E}$, the domain adaptation view remains fully expressive. Given a triple $(P_X, \boldsymbol{\eta}, \widetilde{\boldsymbol{\eta}})$, we can always define a noise transition matrix as:

$$\boldsymbol{E}(x) = \mathbf{1}\widetilde{\boldsymbol{\eta}}^\top,$$

where $\mathbf{1} = [1 \dots 1]^\top$. We can verify that $\boldsymbol{E}$ is row-stochastic, and

$$\widetilde{\boldsymbol{\eta}} = \boldsymbol{E}(x)^\top \boldsymbol{\eta} = (\widetilde{\boldsymbol{\eta}}\mathbf{1}^\top)\boldsymbol{\eta} = \widetilde{\boldsymbol{\eta}}(\mathbf{1}^\top \boldsymbol{\eta}) = \widetilde{\boldsymbol{\eta}}.$$

Therefore, these two perspectives are equivalent.

## A.2    Proofs

### A.2.1    Proof of Proposition 1

**Proposition**

$$\mathcal{A}_0(\boldsymbol{\eta}, \widetilde{\boldsymbol{\eta}}) = \left\{ x \in \mathcal{X} : \arg\max \widetilde{\boldsymbol{\eta}}(x) \subseteq \arg\max \boldsymbol{\eta}(x) \right\}.$$

*Proof.* Notice that

$$\mathcal{M}(x; \boldsymbol{\eta}, \widetilde{\boldsymbol{\eta}}) = 0 \quad \Longleftrightarrow \quad \arg\max \widetilde{\boldsymbol{\eta}}(x) \not\subseteq \arg\max \boldsymbol{\eta}(x).$$

This is because $\mathcal{M}(x; \boldsymbol{\eta}, \widetilde{\boldsymbol{\eta}}) = 0$ when the numerator is zero and the denominator is non-zero, which happens when $\arg\max \widetilde{\boldsymbol{\eta}}(x) \not\subseteq \arg\max \boldsymbol{\eta}(x)$.

An equivalent statement of this is

$$\mathcal{M}(x; \boldsymbol{\eta}, \widetilde{\boldsymbol{\eta}}) > 0 \quad \Longleftrightarrow \quad \arg\max \widetilde{\boldsymbol{\eta}}(x) \subseteq \arg\max \boldsymbol{\eta}(x).$$

∎

### A.2.2    Proof of Proposition 2

**Proposition**    *In the binary setting, for $\kappa \geq 0$,*

$$\mathcal{M}(x; \eta, \widetilde{\eta}) = \max\left\{ \frac{\widetilde{\eta}(x) - \frac{1}{2}}{\eta(x) - \frac{1}{2}}, 0 \right\}, \qquad \textit{and} \qquad \mathcal{A}_\kappa(\eta, \widetilde{\eta}) = \left\{ x \in \mathcal{X} : \frac{\widetilde{\eta}(x) - \frac{1}{2}}{\eta(x) - \frac{1}{2}} > \kappa \right\}.$$

*Proof.* In a brute-force way, we can examine the nine cases where $\eta(x), \widetilde{\eta}(x)$ is greater, equal, or smaller than $1/2$.

If $\widetilde{\eta}(x) > \frac{1}{2}, \eta(x) > \frac{1}{2}$, then

$$
\begin{aligned}
\mathcal{M}(x; \boldsymbol{\eta}, \widetilde{\boldsymbol{\eta}}) &= \min_{j \in \mathcal{Y}} \frac{\max_i [\widetilde{\boldsymbol{\eta}}(x)]_i - [\widetilde{\boldsymbol{\eta}}(x)]_j}{\max_i [\boldsymbol{\eta}(x)]_i - [\boldsymbol{\eta}(x)]_j} \\
&= \frac{\widetilde{\eta}(x) - (1 - \widetilde{\eta}(x))}{\eta(x) - (1 - \eta(x))} \\
&= \frac{\widetilde{\eta}(x) - \frac{1}{2}}{\eta(x) - \frac{1}{2}} \\
&= \max \left\{ \frac{\widetilde{\eta}(x) - \frac{1}{2}}{\eta(x) - \frac{1}{2}}, 0 \right\}.
\end{aligned}
$$

If $\widetilde{\eta}(x) < \frac{1}{2}, \eta(x) < \frac{1}{2}$, the same argument holds.

If $\widetilde{\eta}(x) > \frac{1}{2}, \eta(x) < \frac{1}{2}$ or $\widetilde{\eta}(x) < \frac{1}{2}, \eta(x) > \frac{1}{2}$, take $j = \arg\max \widetilde{\boldsymbol{\eta}}(x)$, we have

$$
\begin{aligned}
\mathcal{M}(x; \boldsymbol{\eta}, \widetilde{\boldsymbol{\eta}}) &= \min_{j \in \mathcal{Y}} \frac{\max_i [\widetilde{\boldsymbol{\eta}}(x)]_i - [\widetilde{\boldsymbol{\eta}}(x)]_j}{\max_i [\boldsymbol{\eta}(x)]_i - [\boldsymbol{\eta}(x)]_j} \\
&= 0 \\
&= \max \left\{ \frac{\widetilde{\eta}(x) - \frac{1}{2}}{\eta(x) - \frac{1}{2}}, 0 \right\}.
\end{aligned}
$$

If $\widetilde{\eta}(x) = \frac{1}{2}, \eta(x) < \frac{1}{2}$ or $\widetilde{\eta}(x) = \frac{1}{2}, \eta(x) < \frac{1}{2}$, take $j \neq \arg\max \boldsymbol{\eta}(x)$, we have

$$
\begin{aligned}
\mathcal{M}(x; \boldsymbol{\eta}, \widetilde{\boldsymbol{\eta}}) &= \min_{j \in \mathcal{Y}} \frac{\max_i [\widetilde{\boldsymbol{\eta}}(x)]_i - [\widetilde{\boldsymbol{\eta}}(x)]_j}{\max_i [\boldsymbol{\eta}(x)]_i - [\boldsymbol{\eta}(x)]_j} \\
&= 0 \\
&= \max \left\{ \frac{\widetilde{\eta}(x) - \frac{1}{2}}{\eta(x) - \frac{1}{2}}, 0 \right\}.
\end{aligned}
$$

If $\eta(x) = \frac{1}{2}$, then

$$
\begin{aligned}
\mathcal{M}(x; \boldsymbol{\eta}, \widetilde{\boldsymbol{\eta}}) &= \min_{j \in \mathcal{Y}} \frac{\max_i [\widetilde{\boldsymbol{\eta}}(x)]_i - [\widetilde{\boldsymbol{\eta}}(x)]_j}{\max_i [\boldsymbol{\eta}(x)]_i - [\boldsymbol{\eta}(x)]_j} \\
&= \frac{\max_i [\widetilde{\boldsymbol{\eta}}(x)]_i - [\widetilde{\boldsymbol{\eta}}(x)]_j}{0} \qquad\qquad \forall j \\
&= +\infty \\
&= \max \left\{ \frac{\widetilde{\eta}(x) - \frac{1}{2}}{\eta(x) - \frac{1}{2}}, 0 \right\}.
\end{aligned}
$$

Note that it makes sense for RSS to be $+\infty$ when $\eta(x) = \frac{1}{2}$, because in this case, clean excess risk $R(f) - R(f^*)$ is 0 at point $x$ for any classifier.

Therefore, we can conclude that

$$
\mathcal{M}(x; \eta, \widetilde{\eta}) = \max \left\{ \frac{\widetilde{\eta}(x) - \frac{1}{2}}{\eta(x) - \frac{1}{2}}, 0 \right\},
$$

by definition, we have, for $\kappa \geq 0$

$$
\begin{aligned}
\mathcal{A}_\kappa(\eta, \widetilde{\eta}) &= \{ x \in \mathcal{X} : \mathcal{M}(x; \eta, \widetilde{\eta}) > \kappa \} \\
&= \left\{ x \in \mathcal{X} : \frac{\widetilde{\eta}(x) - \frac{1}{2}}{\eta(x) - \frac{1}{2}} > \kappa \right\}.
\end{aligned}
$$

$\blacksquare$

### A.2.3 Proof of lower bound: Theorem 1

Now we provide a more formal statement of the minimax lower bound and its proof. We begin with the scenario where the noisy distribution $P_{X\widetilde{Y}}$ has zero Bayes risk as an introductory example. The proof for the general case follows a similar strategy but involves more complex bounding techniques. We recommend that interested readers first review the proof of the zero-error version to build a solid understanding before tackling the general case.

Now consider a more restricted subset of $\Pi(\epsilon, \kappa)$:

$$\Pi(\epsilon, \kappa, V, 0) := \left\{ (P_X, \boldsymbol{\eta}, \widetilde{\boldsymbol{\eta}}) : P_X\left(\mathcal{A}_\kappa\left(\boldsymbol{\eta}, \widetilde{\boldsymbol{\eta}}\right)\right) \geq 1 - \epsilon, P_X \text{ supported on } V + 1 \text{ points}, \widetilde{R}^* = 0 \right\}.$$

**Theorem (Minimax Lower Bound: when $\widetilde{R}^* = 0$)** *Let $\epsilon \in [0, 1], \kappa > 0, V > 1$. For any learning rule $\hat{f}$ based upon $Z^n = \left\{(X_i, \widetilde{Y}_i)\right\}_{i=1}^n$, and $n > \max(V - 1, 2)$,*

$$\sup_{(P_X, \boldsymbol{\eta}, \widetilde{\boldsymbol{\eta}}) \in \Pi(\epsilon, \kappa)} \mathbb{E}_{Z^n}\left[R\left(\hat{f}\right) - R(f^*)\right] \geq \sup_{(P_X, \boldsymbol{\eta}, \widetilde{\boldsymbol{\eta}}) \in \Pi(\epsilon, \kappa, V, 0)} \mathbb{E}_{Z^n}\left[R\left(\hat{f}\right) - R(f^*)\right]$$

$$\geq \frac{K-1}{K}\epsilon + \frac{1}{\kappa}\frac{(V-1)(1-\epsilon)}{8en}$$

*Proof.*

We will construct a triple $(P_X, \boldsymbol{\eta}, \widetilde{\boldsymbol{\eta}})$ that is parameterized by $j, \boldsymbol{b} := [b_1\ b_2\ \cdots\ b_{V-1}]^\top$, and $\delta$.

First, we define $P_X$. Pick any $V + 1$ distinct points $x_0, x_1, \ldots, x_V$,

$$P_X(x) = \begin{cases} \epsilon & x = x_0 \\ (1-\epsilon) \cdot \frac{1}{n} & x = x_1, \ldots, x_{V-1} \\ (1-\epsilon) \cdot \left(1 - \frac{V-1}{n}\right) & x = x_V. \end{cases}$$

this is where we need the condition that $n > V - 1$.

Then, define the clean and noisy class posteriors:

If $x = x_0$, then $\boldsymbol{\eta}(x) = \boldsymbol{e}_j$, $\widetilde{\boldsymbol{\eta}}(x) = \boldsymbol{e}_1$, $\quad j \in \{1, 2, \ldots K\}$ $\qquad\qquad$ (2)

If $x = x_t, 1 \leq t \leq V - 1$, then $\boldsymbol{\eta}(x) = \begin{bmatrix} \frac{1}{2} + \frac{1}{2(\kappa+\delta)} \cdot (-1)^{b_t+1} \\ \frac{1}{2} - \frac{1}{2(\kappa+\delta)} \cdot (-1)^{b_t+1} \\ 0 \\ \vdots \\ 0 \end{bmatrix}$, $\widetilde{\boldsymbol{\eta}}(x) = \boldsymbol{e}_{b_t}, b_t \in \{1, 2\}, \delta > 0,$

$\qquad\qquad$ (3)

If $x = x_V$, then $\boldsymbol{\eta}(x) = \begin{bmatrix} \frac{1}{2} + \frac{1}{2(\kappa+\delta)} \\ \frac{1}{2} - \frac{1}{2(\kappa+\delta)} \\ 0 \\ \vdots \\ 0 \end{bmatrix}$, $\widetilde{\boldsymbol{\eta}}(x) = \boldsymbol{e}_1,$ $\qquad\qquad$ (4)

where $\boldsymbol{e}_i$ denotes the one-hot vector whose $i$-th element is one.

The triple $(P_X, \boldsymbol{\eta}, \widetilde{\boldsymbol{\eta}})$ is thus parameterized by $j, \boldsymbol{b} := [b_1\ b_2\ \cdots\ b_{V-1}]^\top$, and $\delta$.

This construction ensures $(P_X, \boldsymbol{\eta}, \widetilde{\boldsymbol{\eta}}) \in \Pi(\epsilon, \kappa, V, 0)$. In particular,

$$\mathcal{A}_\kappa \supseteq \{x_1, x_2, \ldots, x_V\}, \qquad\qquad P_X(\mathcal{A}_\kappa) \geq 1 - \epsilon,$$
$$\mathcal{X} \setminus \mathcal{A}_\kappa \subseteq \{x_0\}, \qquad\qquad P_X(\mathcal{X} \setminus \mathcal{A}_\kappa) \leq \epsilon,$$

and $\widetilde{R}^* = 0$ because $\widetilde{\boldsymbol{\eta}}(x)$ is one-hot for all $x$.

For any classifier $f$, by definition, its risk equals

$$
\begin{aligned}
R\left(f\right) &= \mathbb{E}_{X,Y}\left[\mathbb{1}_{f(X)\neq Y}\right] \\
&= \mathbb{E}_X \mathbb{E}_{Y|X}[\mathbb{1}_{f(X)\neq Y}] \\
&= \mathbb{E}_X \mathbb{E}_{Y|X}[1 - \mathbb{1}_{f(X)=Y}] \\
&= \mathbb{E}_X\left[1 - [\boldsymbol{\eta}(X)]_{f(X)}\right] \\
&= \int_{\mathcal{X}}\left(1 - [\boldsymbol{\eta}(x)]_{f(x)}\right) dP_X(x).
\end{aligned}
$$

Therefore, the Bayes risk and excess risk equal

$$
\begin{aligned}
R(f^*) &= \inf_f \mathbb{E}_{X,Y}\left[\mathbb{1}_{f(X)\neq Y}\right] \\
&= \int_{\mathcal{X}}\left(1 - \max\boldsymbol{\eta}(x)\right) dP_X(x), \\
R(f) - R(f^*) &= \int_{\mathcal{X}}\left(\max\boldsymbol{\eta}(x) - [\boldsymbol{\eta}(x)]_{f(x)}\right) dP_X(x).
\end{aligned}
$$

Under our construction of $P_X$, $R(f)$ can be decomposed into two parts

$$
R\left(f\right) = \underbrace{\int_{\{x_0\}}\left(1 - [\boldsymbol{\eta}(x)]_{f(x)}\right) dP_X(x)}_{:=R_0(f)} + \underbrace{\int_{\{x_1,\dots,x_V\}}\left(1 - [\boldsymbol{\eta}(x)]_{f(x)}\right) dP_X(x)}_{:=R_V(f)},
$$

and so can the excess risk

$$
\begin{aligned}
R(f) - R(f^*) &= \left(R_0\left(f\right) - R_0(f^*)\right) + \left(R_V\left(f\right) - R_V(f^*)\right) \\
&= \int_{\{x_0\}}\left(\max\boldsymbol{\eta}(x) - [\boldsymbol{\eta}(x)]_{f(x)}\right) dP_X(x) \\
&\quad + \int_{\{x_1,\dots,x_V\}}\left(\max\boldsymbol{\eta}(x) - [\boldsymbol{\eta}(x)]_{f(x)}\right) dP_X(x).
\end{aligned}
$$

Recall that in our construction, $(P_X, \boldsymbol{\eta}, \widetilde{\boldsymbol{\eta}})$ is parameterized by $j, \boldsymbol{b}$, and $\delta$. Therefore

$$
\begin{aligned}
\sup_{(P_X,\boldsymbol{\eta},\widetilde{\boldsymbol{\eta}})\in\Pi(\epsilon,\kappa,V,0)} \mathbb{E}_{Z^n}\left[R\left(\hat{f}\right) - R(f^*)\right] &\geq \sup_{j,\boldsymbol{b},\delta} \mathbb{E}_{Z^n}\left[R\left(\hat{f}\right) - R(f^*)\right] \\
&= \sup_{j,\boldsymbol{b},\delta}\left\{\mathbb{E}_{Z^n}\left[R_0\left(\hat{f}\right) - R_0(f^*)\right]\right. \\
&\qquad\qquad \left. + \mathbb{E}_{Z^n}\left[R_V\left(\hat{f}\right) - R_V(f^*)\right]\right\} \\
&= \sup_j \mathbb{E}_{Z^n}\left[R_0\left(\hat{f}\right) - R_0(f^*)\right] \\
&\qquad + \sup_{\boldsymbol{b},\delta} \mathbb{E}_{Z^n}\left[R_V\left(\hat{f}\right) - R_V(f^*)\right]
\end{aligned}
$$

where the last equality holds because region $\{x_0\}$ only depends on $j$, while region $\{x_1,\dots,x_V\}$ only depends on $\boldsymbol{b}, \delta$.

In the remaining part of the proof, we will examine

$$
\sup_j \mathbb{E}_{Z^n}\left[R_0\left(\hat{f}\right) - R_0(f^*)\right] \tag{5}
$$

and

$$
\sup_{\boldsymbol{b},\delta} \mathbb{E}_{Z^n}\left[R_V\left(\hat{f}\right) - R_V(f^*)\right] \tag{6}
$$

separately.

Let's start with the first term (5), which reflects the excess risk over the "low signal strength" region $\{x_0\}$. Since $\boldsymbol{\eta}$ is one-hot on $\{x_0\}$, its Bayes risk over that is zero

$$\sup_j \mathbb{E}_{Z^n}\left[R_0\left(\hat{f}\right) - R_0(f^*)\right] = \sup_j \mathbb{E}_{Z^n}\left[R_0\left(\hat{f}\right)\right]$$

$$= \sup_j \mathbb{E}_{Z^n}\left[\int_{\{x_0\}} \mathbb{1}_{\hat{f}(x)\neq j} dP_X(x)\right].$$

To deal with $\sup_j$, we use a technique called "the probabilistic method" [Devroye et al., 1996]: replace $j$ with a random variable $J \sim \text{Uniform}\{1, 2, \dots, K\}$:

$$\sup_j \mathbb{E}_{Z^n}\left[\int_{\{x_0\}} \mathbb{1}_{\hat{f}(x)\neq j} dP_X(x)\right] \geq \mathbb{E}_J\left[\mathbb{E}_{Z^n|J}\left[\int_{\{x_0\}} \mathbb{1}_{\hat{f}(x)\neq J} dP_X(x)\right]\right]$$

$$= \mathbb{E}_{J,\, Z^n}\left[\int_{\{x_0\}} \mathbb{1}_{\hat{f}(x)\neq J} dP_X(x)\right]$$

$$= \mathbb{E}_{Z^n}\left[\mathbb{E}_{J|Z^n}\left[\int_{\{x_0\}} \mathbb{1}_{\hat{f}(x)\neq J} dP_X(x)\right]\right].$$

Again, notice that $J$ is an independent draw. Even if the point $x_0$ is observed in $Z^n$, the associated noisy label $\widetilde{Y} = 1$ does not give any information about the clean label $Y = J$. Thus

$$\mathbb{E}_{Z^n}\left[\mathbb{E}_{J|Z^n}\left[\int_{\{x_0\}} \mathbb{1}_{\hat{f}(x)\neq J} dP_X(x)\right]\right] = \mathbb{E}_{Z^n}\left[\mathbb{E}_J\left[\int_{\{x_0\}} \mathbb{1}_{\hat{f}(x)\neq J} dP_X(x)\right]\right]$$

$$= \mathbb{E}_{Z^n}\left[\int_{\{x_0\}} \mathbb{E}_J\left[\mathbb{1}_{\hat{f}(x)\neq J}\right] dP_X(x)\right]$$

$$= \mathbb{E}_{Z^n}\left[\int_{\{x_0\}} \left(1 - \frac{1}{K}\right) dP_X(x)\right]$$

$$= \left(1 - \frac{1}{K}\right)\epsilon.$$

Now we have the minimax lower bound for the first part (5):

$$\sup_j \mathbb{E}_{Z^n}\left[R_{\{x_0\}}\left(\hat{f}\right) - R_{\{x_0\}}(f^*)\right] \geq \left(1 - \frac{1}{K}\right)\epsilon.$$

For the second part (6), which is over $\{x_1, \dots, x_V\}$, due to the relative signal strength condition, and from our explicit construction in Eqn. (3) and (4), the excess risks w.r.t. true and noisy distribution are related by

$$R_V(f) - R_V(f^*) = \int_{\{x_1,\dots,x_V\}} \left(\max \boldsymbol{\eta}(x) - [\boldsymbol{\eta}(x)]_{f(x)}\right) dP_X(x)$$

$$= \int_{\{x_1,\dots,x_V\}} \frac{1}{\kappa + \delta} \left(\max \widetilde{\boldsymbol{\eta}}(x) - [\widetilde{\boldsymbol{\eta}}(x)]_{f(x)}\right) dP_X(x) \quad \text{by construction of } \boldsymbol{\eta}, \widetilde{\boldsymbol{\eta}}$$

$$= \frac{1}{\kappa + \delta} \left(\widetilde{R}_V(f) - \widetilde{R}_V(\tilde{f}^*)\right),$$

where $\widetilde{R}_V(f) := \int_{\{x_1,\dots,x_V\}} \left(1 - [\widetilde{\boldsymbol{\eta}}(x)]_{f(x)}\right) dP_X(x)$. Also note that $f^*(x) = \tilde{f}^*(x)$ for $x \in \{x_1, \dots, x_V\}$, which is a result of our construction of $\boldsymbol{\eta}, \widetilde{\boldsymbol{\eta}}$.

Then

$$\sup_{\boldsymbol{b},\delta} \mathbb{E}_{Z^n}\left[R_V\left(\hat{f}\right) - R_V(f^*)\right] = \sup_{\boldsymbol{b},\delta} \mathbb{E}_{Z^n}\left[\frac{1}{\kappa + \delta}\left(\widetilde{R}_V(f) - \widetilde{R}_V(\tilde{f}^*)\right)\right].$$

This allows us to reduce the label noise problem to a standard learning problem: we have an iid sample $Z^n$ from $P_{X\widetilde{Y}}$ and consider the risk evaluated on the same distribution $P_{X\widetilde{Y}}$. The remainder of the proof is similar to the proof of Devroye et al. [1996, Theorem 14.1].

Notice that by our construction, $\widetilde{Y}$ is a deterministic function of $X$. To be specific, $\widetilde{Y} = \tilde{f}^*(X)$, where

$$\tilde{f}^*(x) = \begin{cases} 1 & x = x_0, \\ b_t & x = x_t, \ 1 \le t \le V - 1 \\ 1 & x = x_V \end{cases}$$

is the noisy Bayes classifier.

We use the shorthand $f_{\boldsymbol{b}} := \tilde{f}^*$ to denote that the noisy Bayes classifier depends on $\boldsymbol{b}$.

Since the noisy Bayes risk is zero,

$$\sup_{\boldsymbol{b},\delta} \ \mathbb{E}_{Z^n} \left[ \frac{1}{\kappa + \delta} \left( \widetilde{R}_V(\hat{f}) - \widetilde{R}_V(\tilde{f}^*) \right) \right] = \sup_{\boldsymbol{b},\delta} \ \frac{1}{\kappa + \delta} \ \mathbb{E}_{Z^n} \left[ \widetilde{R}_V(\hat{f}) \right].$$

Again, use the probabilistic method, replace $\boldsymbol{b}$ with $\boldsymbol{B} \sim \mathrm{Uniform}\{1,2\}^{V-1}$,

$$\sup_{\boldsymbol{b},\delta} \ \frac{1}{\kappa + \delta} \ \mathbb{E}_{Z^n} \left[ \widetilde{R}_V(\hat{f}) \right] \ge \sup_{\delta} \ \frac{1}{\kappa + \delta} \ \mathbb{E}_{\boldsymbol{B},Z^n} \left[ \widetilde{R}_V(\hat{f}) \right]$$

$$= \sup_{\delta} \ \frac{1}{\kappa + \delta} \ \mathbb{E}_{Z^n} \left[ \mathbb{E}_{\boldsymbol{B}|Z^n} \left[ \int_{\{x_1,\ldots,x_V\}} \mathbb{1}_{\hat{f}(x) \ne f_{\boldsymbol{B}}(x)} dP_X(x) \right] \right]$$

Since we have $\boldsymbol{B} \sim \mathrm{Uniform}\{1,2\}^{V-1}$ and also $Z^n|\boldsymbol{B} \sim P_{X\widetilde{Y}}^{\otimes n}$, then by Bayes rule (or eye-balling, since $\widetilde{\boldsymbol{\eta}}$ is one-hot), we get the posterior distribution of $\boldsymbol{B}|Z^n$, to be specific: $\forall x \in \{x_1, \cdots, x_V\}$,

$$\begin{aligned} \text{If } x = X_i, i \in \{1, 2, \ldots, n\}, & \quad \text{then } \mathbb{P}\left(f_{\boldsymbol{B}}(x) = \widetilde{Y}_i | Z^n\right) = 1, \ \mathbb{P}\left(f_{\boldsymbol{B}}(x) \ne \widetilde{Y}_i | Z^n\right) = 0 \\ \text{If } x = x_V, & \quad \text{then } \mathbb{P}\left(f_{\boldsymbol{B}}(x) = 1 | Z^n\right) = 1, \ \mathbb{P}\left(f_{\boldsymbol{B}}(x) = 2 | Z^n\right) = 0 \\ \text{If } x \notin \{X_1, \ldots, X_n, x_V\}, & \quad \text{then } \mathbb{P}\left(f_{\boldsymbol{B}}(x) = 1 | Z^n\right) = \tfrac{1}{2}, \ \mathbb{P}\left(f_{\boldsymbol{B}}(x) = 2 | Z^n\right) = \tfrac{1}{2}, \end{aligned}$$

where we overload the notation $\mathbb{P}$ to denote conditional probability of $\boldsymbol{B}|Z^n$.

Then the optimal decision rule for predicting $\boldsymbol{B}$ based on sample $Z^n$ is:

$$g(x; Z^n) = \begin{cases} \widetilde{Y}_i & x = X_i, i \in \{1, 2, \ldots, n\} \\ 1 & x = x_V \\ \text{random guess from } \{1,2\} & x \ne X_1, \ldots, x \ne X_n, x \ne x_V. \end{cases}$$

Therefore, the error comes from the probability of $X \in \{x_1, \ldots, x_V\}$ not being one of the observed $X_i$: for any $\hat{f}$,

$$
\begin{aligned}
\mathbb{E}_{\boldsymbol{B}, Z^n}\left[\widetilde{R}_V(\hat{f})\right] = \mathbb{E}_{Z^n}\left[\mathbb{E}_{\boldsymbol{B}|Z^n}\left[\int_{\{x_1, \ldots, x_V\}} \mathbb{1}_{\hat{f}(x) \neq f_{\boldsymbol{B}}(x)} dP_X(x)\right]\right] \\
\geq \mathbb{P}\left(X \in \{x_1, \ldots, x_V\}, g(X; Z^n) \neq f_{\boldsymbol{B}}(X)\right) \qquad \because \text{error of } \hat{f} \geq \text{error of } g \\
= \left(1 - \frac{1}{2}\right) \mathbb{P}\left(X \neq X_1, \ldots, X \neq X_n, X \neq x_V, X \in \{x_1, \ldots, x_V\}\right) \\
= \frac{1}{2} \sum_{t=1}^{V} \mathbb{P}\left(X \neq X_1, \ldots, X \neq X_n, X \neq x_V, X = x_t\right) \\
= \frac{1}{2} \sum_{t=1}^{V} \mathbb{P}\left(X_1 \neq x_t, \ldots, X_n \neq x_t, x_V \neq x_t, X = x_t\right) \quad \because \text{replace all } X \text{ with } x_t \\
= \frac{1}{2} \sum_{t=1}^{V-1} \mathbb{P}\left(X_1 \neq x_t, \ldots, X_n \neq x_t, X = x_t\right) \\
= \frac{1}{2} \sum_{t=1}^{V-1} \mathbb{P}\left(X_1 \neq x_t, \ldots, X_n \neq x_t | X = x_t\right) \mathbb{P}\left(X = x_t\right) \\
= \frac{1}{2} \sum_{t=1}^{V-1} \left(1 - \mathbb{P}\left(X = x_t\right)\right)^n \mathbb{P}\left(X = x_t\right) \\
= \frac{1}{2}(V-1)\left(1 - \frac{1-\epsilon}{n}\right)^n \left(\frac{1-\epsilon}{n}\right) \\
= \frac{(V-1)(1-\epsilon)}{2n}\left(1 - \frac{1-\epsilon}{n}\right)^n \\
= \frac{(V-1)(1-\epsilon)}{2n}\left(1 - \frac{1-\epsilon}{n}\right)^{1+\epsilon}\left(1 - \frac{1-\epsilon}{n}\right)^{n-1-\epsilon} \\
\geq \frac{(V-1)(1-\epsilon)}{2n}\left(1 - \frac{1-\epsilon}{n}\right)^{1+\epsilon} e^{-1+\epsilon} \qquad \because \left(1 - \frac{1-\epsilon}{n}\right)^{n-1-\epsilon} \downarrow e^{-1+\epsilon} \\
\geq \frac{(V-1)(1-\epsilon)}{2n}\left(1 - \frac{1}{n}\right)^2 e^{-1} \qquad \because \epsilon \in [0,1] \\
\geq \frac{(V-1)(1-\epsilon)}{2n} \frac{e^{-1}}{4} = \frac{(V-1)(1-\epsilon)}{8en} \qquad \text{take } n > 2.
\end{aligned}
$$

Now we get the minimax risk for the second part (6)

$$
\begin{aligned}
\sup_{\boldsymbol{b}, \delta} \mathbb{E}_{Z^n}\left[R_{\mathcal{A}_\kappa}\left(\hat{f}\right) - R_{\mathcal{A}_\kappa}(f^*)\right] \geq \sup_{\delta} \frac{1}{\kappa + \delta} \frac{(V-1)(1-\epsilon)}{8en} \\
\geq \frac{1}{\kappa} \frac{(V-1)(1-\epsilon)}{8en} \qquad \text{let } \delta \downarrow 0
\end{aligned}
$$

Combine the two parts together, we get the final result, for $n > \max(V - 1, 2)$

$$
\sup_{(P_X, \boldsymbol{\eta}, \widetilde{\boldsymbol{\eta}}) \in \Pi(\epsilon, \kappa, V, 0)} \mathbb{E}_{Z^n}\left[R\left(\hat{f}\right) - R(f^*)\right] \geq \frac{K-1}{K}\epsilon + \frac{1}{\kappa}\frac{(V-1)(1-\epsilon)}{8en}.
$$

∎

As for the general version of the lower bound, now consider the set of triples:

$$\Pi(\epsilon, \kappa, V, L) := \Big\{ (P_X, \boldsymbol{\eta}, \widetilde{\boldsymbol{\eta}}) : P_X\Big(\mathcal{A}_\kappa\left(\boldsymbol{\eta}, \widetilde{\boldsymbol{\eta}}\right)\Big) \geq 1 - \epsilon,$$

$$P_X \text{ supported on } V + 1 \text{ points}, \frac{\widetilde{R}_{\mathcal{A}_\kappa}\left(\tilde{f}^*\right)}{P_X\Big(\mathcal{A}_\kappa\left(\boldsymbol{\eta}, \widetilde{\boldsymbol{\eta}}\right)\Big)} \leq L \Big\},$$

where $\widetilde{R}_C(f) = \int_C \left(1 - [\widetilde{\boldsymbol{\eta}}(x)]_{f(x)}\right) dP_X(x)$.

**Theorem (Minimax Lower Bound (General Version))**  *Let $\epsilon \in [0, 1], \kappa > 0, V > 1, L \in (0, 1/2)$. For any learning rule $\hat{f}$ based upon $Z^n = \{(X_i, \widetilde{Y}_i)\}_{i=1}^n$, for $n \geq \frac{V-1}{2L} \max\left\{16, \frac{1}{(1-2L)^2}\right\}$*

$$\sup_{(P_X, \widetilde{\boldsymbol{\eta}}) \in \Pi(\epsilon, \kappa)} \mathbb{E}_{Z^n}\left[R\left(\hat{f}\right) - R(f^*)\right] \geq \sup_{(P_X, \widetilde{\boldsymbol{\eta}}) \in \Pi(\epsilon, \kappa, V, L)} \mathbb{E}_{Z^n}\left[R\left(\hat{f}\right) - R(f^*)\right]$$

$$\geq \frac{K-1}{K}\epsilon + \frac{1-\epsilon}{\kappa}\sqrt{\frac{(V-1)L}{2n}}e^{-7}$$

$$= \frac{K-1}{K}\epsilon + \Omega\left(\frac{1}{\kappa}\sqrt{\frac{1}{n}}\right).$$

*Proof.*

Now we construct a triple $(P_X, \boldsymbol{\eta}, \widetilde{\boldsymbol{\eta}})$ that is parameterized by $j, \boldsymbol{b} := [b_1\ b_2\ \cdots\ b_{V-1}]^\top, \delta, c$ and $p$.

First, we define $P_X$. Pick any $V + 1$ distinct points $x_0, x_1, \ldots, x_V$,

$$P_X(x) = \begin{cases} \epsilon & x = x_0 \\ (1-\epsilon) \cdot p & x = x_1, \ldots, x_{V-1} \\ (1-\epsilon) \cdot (1 - (V-1)p) & x = x_V. \end{cases}$$

This imposes the constraint $(V - 1)p \leq 1$, which will be satisfied in the end. Notice the difference compared to the previous zero-error proof: we place probability mass $p$, rather than $1/n$, on $x_1, \ldots, x_{V-1}$.

As for the clean and noisy class probabilities, choose

If $x = x_0$, then $\boldsymbol{\eta}(x) = \boldsymbol{e}_j, \widetilde{\boldsymbol{\eta}}(x) = \boldsymbol{e}_1, \quad j \in \{1, 2, \ldots k\}$ (7)

If $x = x_t, 1 \leq t \leq V - 1$, then $\boldsymbol{\eta}(x) = \begin{bmatrix} \frac{1}{2} + \frac{c}{\kappa+\delta} \cdot (-1)^{b_t+1} \\ \frac{1}{2} - \frac{c}{\kappa+\delta} \cdot (-1)^{b_t+1} \\ 0 \\ \vdots \\ 0 \end{bmatrix}, \widetilde{\boldsymbol{\eta}}(x) = \begin{bmatrix} \frac{1}{2} + c \cdot (-1)^{b_t+1} \\ \frac{1}{2} - c \cdot (-1)^{b_t+1} \\ 0 \\ \vdots \\ 0 \end{bmatrix},$

$$b_t \in \{1, 2\},\ \delta > 0,\ c \in \left(0, \frac{1}{2}\right)$$ (8)

If $x = x_V$, then $\boldsymbol{\eta}(x) = \begin{bmatrix} \frac{1}{2} + \frac{1}{2(\kappa+\delta)} \\ \frac{1}{2} - \frac{1}{2(\kappa+\delta)} \\ 0 \\ \vdots \\ 0 \end{bmatrix}, \widetilde{\boldsymbol{\eta}}(x) = \boldsymbol{e}_1,$ (9)

where $\boldsymbol{e}_i$ denotes the one-hot vector whose $i$-th element is one.

The construction for class posterior is also similar to the previous proof, except that for $x = x_t, t \in \{1, \ldots, V - 1\}, \widetilde{\boldsymbol{\eta}}$ is no longer a one-hot vector, rather has class probability separated by $2c$: $\left|[\widetilde{\boldsymbol{\eta}}(x)]_1 - [\widetilde{\boldsymbol{\eta}}(x)]_2\right| = 2c$.

Therefore, the triple $(P_X, \boldsymbol{\eta}, \widetilde{\boldsymbol{\eta}})$ can be parameterized by $j, \boldsymbol{b} := [b_1 \ b_2 \ \cdots \ b_{V-1}]^\top$, $\delta$, $c$ and $p$.

Again, this construction ensures $(P_X, \boldsymbol{\eta}, \widetilde{\boldsymbol{\eta}}) \in \Pi(\epsilon, \kappa)$, to be specific:

$$\mathcal{A}_\kappa \supseteq \{x_1, x_2, \ldots, x_V\}, \qquad\qquad P_X(\mathcal{A}_\kappa) \geq 1 - \epsilon,$$
$$\mathcal{X} \setminus \mathcal{A}_\kappa \subseteq \{x_0\}, \qquad\qquad P_X(\mathcal{X} \setminus \mathcal{A}_\kappa) \leq \epsilon.$$

For any classifier $f$, its risk can be decomposed into two parts

$$R(f) = \underbrace{\int_{\{x_0\}} \left(1 - [\boldsymbol{\eta}(x)]_{f(x)}\right) dP_X(x)}_{:=R_0(f)} + \underbrace{\int_{\{x_1, \ldots, x_V\}} \left(1 - [\boldsymbol{\eta}(x)]_{f(x)}\right) dP_X(x)}_{:=R_V(f)},$$

as can its excess risk

$$R(f) - R(f^*) = \left(R_0(f) - R_0(f^*)\right) + \left(R_V(f) - R_V(f^*)\right).$$

In our construction, $(P_X, \boldsymbol{\eta}, \widetilde{\boldsymbol{\eta}})$ is parameterized by $j, \boldsymbol{b} := [b_1 \ b_2 \ \cdots \ b_{V-1}]^\top$, $\delta$, $c$ and $p$, therefore

$$\sup_{(P_X, \boldsymbol{\eta}, \widetilde{\boldsymbol{\eta}}) \in \Pi(\epsilon, \kappa, V, L)} \mathbb{E}_{Z^n} \left[R\left(\hat{f}\right) - R(f^*)\right] \geq \sup_j \mathbb{E}_{Z^n} \left[R_0\left(\hat{f}\right) - R_0(f^*)\right] \tag{10}$$

$$+ \sup_{\boldsymbol{b}, \delta, c, p} \mathbb{E}_{Z^n} \left[\frac{1}{\kappa + \delta} \left(\widetilde{R}_V(f) - \widetilde{R}_V(\tilde{f}^*)\right)\right]. \tag{11}$$

Note that we have used the fact that

$$R_V(f) - R_V(f^*) = \frac{1}{\kappa + \delta} \left(\widetilde{R}_V(f) - \widetilde{R}_V(\tilde{f}^*)\right),$$

where $\widetilde{R}_V(f) := \int_{\{x_1, \ldots, x_V\}} \left(1 - [\widetilde{\boldsymbol{\eta}}(x)]_{f(x)}\right) dP_X(x)$.

The first part (10) is exactly the same as in the zero-error proof, and we have

$$\sup_j \mathbb{E}_{Z^n} \left[R_0\left(\hat{f}\right) - R_0(f^*)\right] \geq \left(1 - \frac{1}{K}\right) \epsilon.$$

From this point forward, the procedure is similar to the proof of Devroye et al. [1996, Theorem 14.5]. For the second part (11), the noisy Bayes classifier is still

$$\tilde{f}^*(x) = \begin{cases} j & x = x_0, \\ b_t & x = x_t, \ 1 \leq t \leq V \\ 1 & x = x_V. \end{cases}$$

We also use the shorthand $f_{\boldsymbol{b}} := \tilde{f}^*$ to denote that the noisy Bayes classifier depends on $\boldsymbol{b}$.

Now the noisy Bayes risk is no longer zero. In fact

$$\widetilde{R}_V(\tilde{f}^*) = \int_{\{x_1, \ldots, x_V\}} \left(1 - [\widetilde{\boldsymbol{\eta}}(x)]_{f(x)}\right) dP_X(x) = (V-1)(1-\epsilon)p \left(\frac{1}{2} - c\right)$$

What's more,

$$\frac{\widetilde{R}_{\mathcal{A}_\kappa}\left(\tilde{f}^*\right)}{P_X\left(\mathcal{A}_\kappa(\boldsymbol{\eta}, \widetilde{\boldsymbol{\eta}})\right)} \leq \frac{\widetilde{R}_V(\tilde{f}^*)}{P_X\left(\{x_1, \ldots, x_V\}\right)} = (V-1)p \left(\frac{1}{2} - c\right), \tag{12}$$

where the inequality holds from $\widetilde{R}_{\mathcal{A}_\kappa}(\tilde{f}^*) = \widetilde{R}_V(\tilde{f}^*)$ (because $\widetilde{\boldsymbol{\eta}}$ is one-hot at point $x_0$) and $P_X\left(\mathcal{A}_\kappa(\boldsymbol{\eta}, \widetilde{\boldsymbol{\eta}})\right) \geq P_X\left(\{x_1, \ldots, x_V\}\right)$.

Notice that in order to ensure that our construction $(P_X, \boldsymbol{\eta}, \widetilde{\boldsymbol{\eta}}) \in \Pi(\epsilon, \kappa, V, L)$, by definition

$$\frac{\widetilde{R}_{\mathcal{A}_\kappa}\left(\tilde{f}^*\right)}{P_X\left(\mathcal{A}_\kappa\left(\boldsymbol{\eta}, \widetilde{\boldsymbol{\eta}}\right)\right)} \leq L,$$

Due to the upper bound of (12), it suffices to require that

$$(V-1)p\left(\frac{1}{2} - c\right) = L, \tag{13}$$

and this ensures that $(P_X, \boldsymbol{\eta}, \widetilde{\boldsymbol{\eta}}) \in \Pi(\epsilon, \kappa, V, L)$ upon recalling that $(P_X, \boldsymbol{\eta}, \widetilde{\boldsymbol{\eta}}) \in \Pi(\epsilon, \kappa)$, and that $P_X$ is supported on $V+1$ points.

It should be noted that since $(V-1)p \leq 1$ is required, and since $c > 0$, we have $L < 1 \cdot 1/2$. This is the origin of our condition $L < 1/2$ in the statement of the theorem. Naturally, the statement can be adjusted to $\min(L, 1/2)$ instead. In any case, we are left with two nontrivial constraint on our parameters $(p, c)$: (13) and $(V-1)p \leq 1$, along with the boundary consraints $p \in [0, 1]$ and $c \in [0, 1/2]$.

For fixed $\boldsymbol{b}$, plugging in the definition of $\widetilde{\boldsymbol{\eta}}$, the excess risk over region $\{x_1, \ldots, x_V\}$ becomes

$$\widetilde{R}_V(\hat{f}) - \widetilde{R}_V(\tilde{f}^*) = \int_{\{x_1, \ldots, x_V\}} 2c\mathbb{1}_{\hat{f}(x) \neq f_{\boldsymbol{b}}(x)} dP_X(x)$$

$$\geq 2c \sum_{t=1}^{V-1} (1-\epsilon)p\mathbb{1}_{\hat{f}(x_t) \neq f_{\boldsymbol{b}}(x_t)},$$

where the inequality follows from the fact that we ignore the risk on point $x_V$.

Using the probabilistic method, replace $\boldsymbol{b}$ with $\boldsymbol{B} \sim \text{Uniform}\{1, 2\}^{V-1}$,

$$\sup_{\boldsymbol{b}, \delta, c, p} \mathbb{E}_{Z^n}\left[\frac{1}{\kappa+\delta}\left(\widetilde{R}_V(\hat{f}) - \widetilde{R}_V(\tilde{f}^*)\right)\right] \geq \sup_{\delta, c, p} \mathbb{E}_{\boldsymbol{B}, Z^n}\left[\frac{1}{\kappa+\delta}\left(\widetilde{R}_V(\hat{f}) - \widetilde{R}_V(\tilde{f}^*)\right)\right]$$

$$= \sup_{\delta, c, p} \frac{1}{\kappa+\delta}\mathbb{E}_{Z^n}\left[\mathbb{E}_{\boldsymbol{B}|Z^n}\left[\left(\widetilde{R}_V(\hat{f}) - \widetilde{R}_V(\tilde{f}^*)\right)\right]\right]$$

Now, we need to calculate $\boldsymbol{B}|Z^n$, which can be calculated using Bayes rule because we have $\boldsymbol{B} \sim \text{Uniform}\{1, 2\}^{V-1}$ and also $Z^n|\boldsymbol{B} \sim P_{X\widetilde{Y}}^{\otimes n}$.

To be specific, for any $x \in \{x_0, x_1, \ldots, x_{V-1}\}$, assume point $x_t$ is observed $k$ times in training sample $Z^n$,

$$\mathbb{P}\left(f_{\boldsymbol{B}}(x) = 1|Z^n\right) = \begin{cases} \frac{1}{2} & x \neq X_1, \ldots, x \neq X_n, x \neq x_V \\ \mathbb{P}\left(B_t = 1|Y_{t_1}, \ldots, Y_{t_k}\right) & x = x_t = X_{t_1} = \cdots = X_{t_k}, 1 \leq t \leq V-1, \end{cases}$$

where $B_t$ denotes the $t$-th element of vector $\boldsymbol{B}$ (that associates with $x_t$).

Next we compute $\mathbb{P}\left(B_t = 1|Y_{t_1} = y_1, \ldots, Y_{t_k} = y_k\right)$ for $y_1, \ldots, y_k \in \{1, 2\}$. Denote the numbers of ones and twos by $k_1 = |\{j \leq k : y_j = 1\}|$ and $k_2 = |\{j \leq k : y_j = 2\}|$. Using Bayes rule, we get

$$\mathbb{P}\left(B_t = 1|Y_{t_1}, \ldots, Y_{t_k}\right) = \frac{\mathbb{P}\left(B_t = 1 \cap Y_{t_1}, \ldots, Y_{t_k}\right)}{\mathbb{P}\left(Y_{t_1}, \ldots, Y_{t_k}\right)}$$

$$= \frac{\mathbb{P}\left(Y_{t_1}, \ldots, Y_{t_k}|B_t = 1\right)\mathbb{P}\left(B_t = 1\right)}{\sum_{i=1}^2 \mathbb{P}\left(Y_{t_1}, \ldots, Y_{t_k}|B_t = i\right)\mathbb{P}\left(B_t = i\right)}$$

$$= \frac{(1/2+c)^{k_1}(1/2-c)^{k_2}(1/2)}{(1/2+c)^{k_1}(1/2-c)^{k_2}(1/2) + (1/2+c)^{k_2}(1/2-c)^{k_1}(1/2)}.$$

After some calculation, following the proof of Devroye et al. [1996, Theorem 14.5], we get

$$\sup_{b,\delta,c,p} \mathbb{E}_{Z^n} \left[ \frac{1}{\kappa + \delta} \left( \widetilde{R}_{\mathcal{A}_\kappa}(f) - \widetilde{R}_{\mathcal{A}_\kappa}(\tilde{f}^*) \right) \right]$$

$$\geq \sup_{\delta,c,p} \frac{1}{\kappa + \delta} c(V-1)(1-\epsilon)p e^{-\frac{8n(1-\epsilon)pc^2}{1-2c} - \frac{4c\sqrt{n(1-\epsilon)p}}{1-2c}}$$

$$\geq \frac{1-\epsilon}{\kappa} \sup_{c,p} c(V-1)p e^{-\frac{8npc^2}{1-2c} - \frac{4c\sqrt{np}}{1-2c}} \quad \because \epsilon \geq 0, \text{ take } \delta \downarrow 0$$

$$= \frac{1-\epsilon}{\kappa} \sup_{c,p} c \frac{L}{1/2 - c} e^{-\frac{8npc^2}{1-2c} - \frac{4c\sqrt{np}}{1-2c}}, \quad \because (13) \tag{14}$$

where the supremum is over $(p,c) \in [0,1] \times [0,1/2]$ such that

$$(V-1)p \leq 1, \text{ and } (V-1)p(1/2 - c) = L.$$

Now, suppose $n$ is so large that

$$n \geq \frac{(V-1)}{8L(1/2 - L)^2} \iff L \leq \frac{1}{2} - \sqrt{\frac{(V-1)}{8nL}},$$

and further that

$$\sqrt{\frac{(V-1)}{8nL}} \leq \frac{1}{8} \iff n \geq \frac{8(V-1)}{L}.$$

We choose

$$c = \sqrt{\frac{(V-1)}{8nL}}, \text{ and } p = \frac{L}{(V-1)(1/2 - c)}.$$

By our choice of $c$ and the first condition on $n$ above, we can conclude that $L \leq (1/2 - c)$, and therefore,

$$(V-1)p = \frac{L}{1/2 - c} \leq 1,$$

meaning that both the constraints required on $(p,c)$ are met by the above choice.

As a consequence of this choice of $c, p$, we observe that

$$npc^2 = \frac{nL}{(V-1)(1/2 - c)} \cdot c^2 = \frac{nL}{(V-1)(1/2 - c)} \cdot \frac{(V-1)}{8nL} = \frac{1}{4 - 8c} \leq \frac{1}{3}.$$

Since $c \leq 1/8$ further implies that $\frac{1}{1-2c} \leq \frac{4}{3}$, this implies that

$$\frac{8npc^2}{1-2c} + \frac{4\sqrt{npc^2}}{1-2c} \leq \frac{8}{3} \cdot \frac{4}{3} + 4 \cdot \frac{4}{3} \cdot \sqrt{\frac{1}{3}} \leq 7.$$

Thus, instantiating the bound (14), we conclude that

$$\sup_{b,\delta,c,p} \mathbb{E}_{Z^n} \left[ \frac{1}{\kappa + \delta} \left( \widetilde{R}_{\mathcal{A}_\kappa}(f) - \widetilde{R}_{\mathcal{A}_\kappa}(\tilde{f}^*) \right) \right] \geq \frac{1-\epsilon}{\kappa} \cdot \sqrt{\frac{V-1}{8nL}} \cdot \frac{L}{1/2 - c} \cdot e^{-7}$$

$$\geq \frac{1-\epsilon}{\kappa} \sqrt{\frac{(V-1)L}{8n}} e^{-7} \cdot 2$$

$$= \frac{1-\epsilon}{\kappa} \sqrt{\frac{(V-1)L}{2n}} e^{-7}.$$

Putting the two parts together

$$\sup_{(P_X,\boldsymbol{\eta},\tilde{\boldsymbol{\eta}}) \in \Pi(\epsilon,\kappa)} \mathbb{E}_{Z^n} \left[ R\left(\hat{f}\right) - R(f^*) \right] \geq \frac{K-1}{K}\epsilon + \frac{1-\epsilon}{\kappa} \sqrt{\frac{(V-1)L}{2n}} e^{-7},$$

for $n \geq \frac{V-1}{2L} \max\left\{ 16, \frac{1}{(1-2L)^2} \right\}$.

$\blacksquare$

### A.2.4 Proof of upper bound: Lemma 1

**Lemma (Oracle Inequality under Feature-dependent Label Noise)** *For any $(P_X, \boldsymbol{\eta}, \widetilde{\boldsymbol{\eta}})$ and any classifier $f$, we have*

$$R(f) - R(f^*) \leq \inf_{\kappa > 0} \left\{ P_X\left(\mathcal{X} \setminus \mathcal{A}_\kappa\left(\boldsymbol{\eta}, \widetilde{\boldsymbol{\eta}}\right)\right) + \frac{1}{\kappa}\left(\widetilde{R}(f) - \widetilde{R}\left(\tilde{f}^*\right)\right) \right\}.$$

*Proof.* For any $\kappa \geq 0$, the input space $\mathcal{X}$ can be divided into two regions: $\mathcal{X} \setminus \mathcal{A}_\kappa$ and $\mathcal{A}_\kappa$.

For any $f$, its risk is

$$
\begin{aligned}
R(f) &= \mathbb{E}_{X,Y}\left[\mathbb{1}_{f(X) \neq Y}\right] \\
&= \mathbb{E}_X \mathbb{E}_{Y|X}[\mathbb{1}_{f(X) \neq Y}] \\
&= \mathbb{E}_X \mathbb{E}_{Y|X}[1 - \mathbb{1}_{f(X) = Y}] \\
&= \mathbb{E}_X\left[1 - [\boldsymbol{\eta}(X)]_{f(X)}\right] \\
&= \int_{\mathcal{X}} \left(1 - [\boldsymbol{\eta}(x)]_{f(x)}\right) dP_X(x).
\end{aligned}
$$

Therefore, its excess risk is

$$
\begin{aligned}
R(f) - R(f^*) &= \int_{\mathcal{X}} \left(\max \boldsymbol{\eta}(x) - [\boldsymbol{\eta}(x)]_{f(x)}\right) dP_X(x) \\
&= \underbrace{\int_{\mathcal{X} \setminus \mathcal{A}_\kappa} \left(\max \boldsymbol{\eta}(x) - [\boldsymbol{\eta}(x)]_{f(x)}\right) dP_X(x)}_{\text{ⓐ}} \\
&\quad + \underbrace{\int_{\mathcal{A}_\kappa} \left(\max \boldsymbol{\eta}(x) - [\boldsymbol{\eta}(x)]_{f(x)}\right) dP_X(x)}_{\text{ⓑ}}
\end{aligned}
$$

Now examine the two terms separately,

$$\text{ⓐ} \leq \int_{\mathcal{X} \setminus \mathcal{A}_\kappa} 1 \, dP_X(x) = P_X\left(\mathcal{X} \setminus \mathcal{A}_\kappa\left(\boldsymbol{\eta}, \widetilde{\boldsymbol{\eta}}\right)\right),$$

and

$$
\begin{aligned}
\text{ⓑ} &< \int_{\mathcal{A}_\kappa} \frac{1}{\kappa}\left(\max \widetilde{\boldsymbol{\eta}}(x) - [\widetilde{\boldsymbol{\eta}}(x)]_{f(x)}\right) dP_X(x) \quad &\because \text{by definition of relative signal strength} \\
&\leq \int_{\mathcal{X}} \frac{1}{\kappa}\left(\max \widetilde{\boldsymbol{\eta}}(x) - [\widetilde{\boldsymbol{\eta}}(x)]_{f(x)}\right) dP_X(x) \\
&= \frac{1}{\kappa}\left(\widetilde{R}(f) - \widetilde{R}(\tilde{f}^*)\right) \quad &\because \text{by definition of } \widetilde{R}.
\end{aligned}
$$

Since this works for any $\kappa > 0$, we then have

$$R(f) - R(f^*) \leq \inf_{\kappa > 0} \left\{ P_X\left(\mathcal{X} \setminus \mathcal{A}_\kappa\left(\boldsymbol{\eta}, \widetilde{\boldsymbol{\eta}}\right)\right) + \frac{1}{\kappa}\left(\widetilde{R}(f) - \widetilde{R}\left(\tilde{f}^*\right)\right) \right\}.$$

∎

### A.2.5 Proof of upper bound: Theorem 2

To set the stage for the rate of convergence proof, we first introduce the concept of shattering in the multiclass setting and the Natarajan dimension [Natarajan, 1989], which serves as a multiclass counterpart to the VC dimension [Vapnik and Chervonenkis, 1971].

**Definition 4 (Multiclass Shattering)** *Let $\mathcal{H}$ be a class of functions from $\mathcal{X}$ to $\mathcal{Y} = \{1, 2, \ldots, K\}$. For any set containing $n$ distinct elements $C_n = \{x_1, \ldots, x_n\} \subset \mathcal{X}$, denote*

$$\mathcal{H}_{C_n} = \{(h(x_1), \ldots, h(x_n)) : h \in \mathcal{H}\},$$

*and therefore $|\mathcal{H}_{C_n}|$ is the number of distinct vectors of length $n$ that can be realized by functions in $\mathcal{H}$.*

*The $n^{th}$ shatter coefficient is defined as*

$$S(\mathcal{H}, n) := \max_{C_n} |\mathcal{H}_{C_n}|.$$

*We say that a set $C_n$ is shattered by $\mathcal{H}$ if there exists $f, g : C_n \to \mathcal{Y}$ such that for every $x \in C_n$, $f(x) \neq g(x)$, and*

$$\mathcal{H}_C \supseteq \{f(x_1), g(x_1)\} \times \{f(x_2), g(x_2)\} \times \cdots \times \{f(x_n), g(x_n)\}$$

If $\mathcal{Y} = \{1, 2\}$, this definition reduces to the binary notion of shattering which says all labeling of points can be realized by some function in the hypothesis class $\mathcal{H}$, i.e., $\mathcal{H}_C = \{1, 2\}^{|C|}$. Note that multiclass shattering does not mean being able to realize all $K$ possible labels for each point $x \in C$. Instead, multiclass shattering is more like "embed the binary cube into multiclass", where every $x \in C$ is allowed to pick from two of the $K$ labels.

**Definition 5 (Natarajan Dimension)** *The Natarajan dimension of $\mathcal{H}$, denoted Ndim($\mathcal{H}$), is the maximal size of a shattered set $C \in \mathcal{X}$.*

**Theorem (Excess Risk Upper Bound of NI-ERM)** *Let $\epsilon \in [0, 1], \kappa \in (0, +\infty)$. Consider any $(P_X, \boldsymbol{\eta}, \widetilde{\boldsymbol{\eta}}) \in \Pi(\epsilon, \kappa)$, assume function class $\mathcal{F}$ has Natarajan dimension $V$, and the noisy Bayes classifier $\tilde{f}^*$ belongs to $\mathcal{F}$. Let $\hat{f} \in \mathcal{F}$ be the ERM trained on $Z^n = \{(X_i, \widetilde{Y}_i)\}_{i=1}^{n}$, then*

$$\mathbb{E}_{Z^n}\left[R\left(\hat{f}\right) - R(f^*)\right] \leq \epsilon + \frac{1}{\kappa} \cdot 16 \sqrt{\frac{V \log n + 2V \log k + 4}{2n}}$$

$$= \epsilon + \mathcal{O}\left(\frac{1}{\kappa}\sqrt{\frac{V}{n}}\right) \quad \text{up to log factor.}$$

*Proof.* Following directly from Lemma 1, with $(P_X, \boldsymbol{\eta}, \widetilde{\boldsymbol{\eta}}) \in \Pi(\epsilon, \kappa)$, we already have

$$R(f) - R(f^*) \leq P_X\left(\mathcal{X} \setminus \mathcal{A}_\kappa\left(\boldsymbol{\eta}, \widetilde{\boldsymbol{\eta}}\right)\right) + \frac{1}{\kappa}\left(\widetilde{R}(f) - \widetilde{R}\left(\tilde{f}^*\right)\right)$$

$$\leq \epsilon + \frac{1}{\kappa}\left(\widetilde{R}(f) - \widetilde{R}\left(\tilde{f}^*\right)\right).$$

Now replace $f$ with NI-ERM $\hat{f}$. To bound the expected excess risk we employ a multiclass VC-style inequality.

**Lemma 2**

$$\mathbb{E}_{Z^n}\left[\widetilde{R}\left(\hat{f}\right) - \widetilde{R}\left(\tilde{f}^*\right)\right] \leq 16 \sqrt{\frac{\log(8eS(\mathcal{H}, n))}{2n}}$$

The binary version of this lemma is Corollary 12.1 in Devroye et al. [1996]. We prove the multiclass version below in Section A.2.6.

Next, we bound the multiclass shattering coefficient with Natarajan dimension, using the following lemma, which can be viewed as a multiclass version of Sauer's lemma.

**Lemma 3 (Natarajan [1989])** *Let $C$ and $\mathcal{Y}$ be two finite sets and let $\mathcal{H}$ be a set of functions from $C$ to $\mathcal{Y}$. Then*

$$|\mathcal{H}| \leq |C|^{Ndim(\mathcal{H})} \cdot |\mathcal{Y}|^{2Ndim(\mathcal{H})}.$$

Letting $V$ denote $\mathrm{Ndim}(\mathcal{H})$, we have that $S(\mathcal{H}, n) \leq n^V K^{2V}$, and therefore Lemma 2 can be upper bounded by

$$
\mathbb{E}_{Z^n}\left[\widetilde{R}\left(\hat{f}\right) - \widetilde{R}\left(\tilde{f}^*\right)\right] \leq 16\sqrt{\frac{\log\left(8e(n)^V K^{2V}\right)}{2n}}
$$
$$
= 16\sqrt{\frac{\log 8e + \log\left(n^V\right) + \log\left(K^{2V}\right)}{2n}}
$$
$$
\leq 16\sqrt{\frac{V\log n + 2V\log K + 4}{2n}}
$$

Putting things together,

$$
\mathbb{E}_{Z^n}\left[R\left(\hat{f}\right) - R(f^*)\right] \leq \epsilon + \frac{1}{\kappa}\cdot 16\sqrt{\frac{V\log n + 2V\log K + 4}{2n}}.
$$

∎

### A.2.6 Proof of Lemma 2

**Theorem 6** *Consider any set of multiclass classifiers $\mathcal{F}$. Let $(X_1, Y_1), \ldots, (X_n, Y_n)$ be iid draws from $P_{XY}$. For any $n$, and any $\epsilon > 0$,*

$$
\Pr\left\{\sup_{f\in\mathcal{F}}|R_n(f) - R(f)| > \epsilon\right\} \leq 8S(\mathcal{F}, n)e^{-n\epsilon^2/32}
$$

*where the probability is with respect to the draw of the data.*

*Proof.* Apply Theorem 12.5 from Devroye et al. [1996], with the following identifications. In what follows, the left-hand side of each equation is a notation from Devroye et al. [1996], and the right-hand side is our notation.

$$
\nu = P_{XY}
$$
$$
Z = (X, Y)
$$
$$
Z_i = (X_i, Y_i)
$$
$$
\mathcal{A} = \{A_f \mid f \in \mathcal{F}\}, \text{ where } A_f := \{(x, y) \mid f(x) = y\}
$$

With these identifications, we have

$$
\nu(A_f) = 1 - R(f)
$$
$$
\nu_n(A_f) = \frac{1}{n}\sum_i \mathbb{1}_{\{Z_i \in A_f\}} = \frac{1}{n}\sum_i \mathbb{1}_{\{f(X_i)=Y_i\}} = 1 - R_n(f)
$$

By Theorem 12.5 we conclude

$$
\Pr\left\{\sup_{f\in\mathcal{F}}|R_n(f) - R(f)| > \epsilon\right\} \leq 8s(\mathcal{A}, n)e^{-n\epsilon^2/32},
$$

where $s(\mathcal{A}, n)$ (note the lowercase "s") is defined to be

$$
\max_{z_1,\ldots,z_n} \mathcal{N}_{\mathcal{A}}(z_1, \ldots, z_n)
$$

where the max is over points $z_1, \ldots, z_n$, and $\mathcal{N}_{\mathcal{A}}(z_1, \ldots, z_n)$ is the number of distinct subsets of the form

$$
A_f \cap \{z_1, \ldots, z_n\}
$$

as $f$ ranges over $\mathcal{F}$.

To conclude the proof, it suffices to show that $s(\mathcal{A}, n) \leq S(\mathcal{F}, n)$, where the latter expression is the multiclass shatter coefficient defined above. We show this as follows.

Consider fixed pairs $z_i = (x_i, y_i)$, $i = 1, \ldots, n$. Supposed that there are $N$ distinct subsets of the form $A_f \cap \{z_1, \ldots, z_n\}$, and let $f_1, \ldots, f_N$ be the classifiers in $\mathcal{F}$ that realize these distinct subsets. Consider the map that sends $f_i$ to the vector of its values at $x_1, \ldots, x_n$:

$$f_i \mapsto (f_i(x_1), \ldots, f_i(x_n)) \in \mathcal{Y}^n.$$

We will show that this map is injective, from which the claim follows. To see injectivity, consider classifiers $f_i$ and $f_j$, where $i \neq j$. Since $f_i$ and $f_j$ yield different subsets, it means there is some pair $(x_k, y_k)$ such that one of $f_i$ and $f_j$ classifies the pair correctly, while the other does not. This implies that $f_i(x_k) \neq f_j(x_k)$, and therefore

$$(f_i(x_1), \ldots, f_i(x_n)) \neq (f_j(x_1), \ldots, f_j(x_n)).$$

This concludes the proof. ∎

Now, Lemma 2 follows from the above theorem (stated in terms of the noisy data/distribution/risk) in precisely the same way that Corollary 12.1 in Devroye et al. [1996] follows from Theorem 12.6 in the same book.

### A.2.7 Proof of upper bound: Theorem 3

**Theorem (Excess Risk Upper Bound of NI-ERM under smooth relative margin condition)** *Let $\epsilon \in [0, 1], \alpha > 0, C_\alpha > 0$. Consider any $(P_X, \boldsymbol{\eta}, \widetilde{\boldsymbol{\eta}}) \in \Pi'(\epsilon, \alpha, C_\alpha)$, assume function class $\mathcal{F}$ has Natarajan dimension $V$, and the noisy Bayes classifier $\tilde{f}^*$ belongs to $\mathcal{F}$. Let $\hat{f} \in \mathcal{F}$ be the ERM trained on $Z^n = \{(X_i, \widetilde{Y}_i)\}_{i=1}^n$. Then*

$$\mathbb{E}_{Z^n}\left[R\left(\hat{f}\right) - R\left(f^*\right)\right] \leq \epsilon + \inf_{\kappa > 0}\left\{C_\alpha \kappa^\alpha + \tilde{\mathcal{O}}\left(\frac{1}{\kappa}\sqrt{\frac{V}{n}}\right)\right\}$$

$$= \epsilon + \tilde{\mathcal{O}}\left(n^{-\alpha/(2+2\alpha)}\right).$$

*Proof.* Again, using Lemma 1, and Theorem 2, we can conclude the following, where $C$ is some large enough constant.

$$\mathbb{E}_{Z^n}[R(\hat{f}) - R(f^*)]$$

$$\leq \inf_{\kappa > 0}\left\{P_X\left(\mathcal{X} \setminus \mathcal{A}_\kappa\left(\boldsymbol{\eta}, \widetilde{\boldsymbol{\eta}}\right)\right) + \frac{1}{\kappa}\left(\widetilde{R}(f) - \widetilde{R}\left(\tilde{f}^*\right)\right)\right\}$$

$$\leq \inf_{\kappa > 0}\left\{P_X\left(\mathcal{X} \setminus \mathcal{A}_\kappa\left(\boldsymbol{\eta}, \widetilde{\boldsymbol{\eta}}\right)\right) + \frac{1}{\kappa}\sqrt{\frac{CV\log(nK)}{n}}\right\}.$$

Now, by definition of $\Pi'(\epsilon, \alpha, C_\alpha)$, it holds that

$$\forall \kappa > 0, \ P_X(\mathcal{M}(x; \boldsymbol{\eta}, \widetilde{\boldsymbol{\eta}}) \leq \kappa) \leq C_\alpha \kappa^\alpha + \epsilon.$$

Thus, we can further conclude that

$$\mathbb{E}_{Z^n}[R(\hat{f}) - R(f^*)] \leq \inf_{\kappa > 0}\left\{\epsilon + C_\alpha \kappa^\alpha + \frac{1}{\kappa}\sqrt{\frac{CV\log(nK)}{n}}\right\}.$$

The final statement now comes from optimizing the above bound, which is attained by taking the derivative w.r.t. $\kappa$ and set to zero, we have

$$\kappa_* = \left((\alpha C_\alpha)^{-1}\sqrt{CV\log(nK)/n}\right)^{1/(\alpha+1)}.$$

This yields the bound

$$\mathbb{E}_{Z^n}[R(\hat{f}) - R(f^*)] \leq \epsilon + \mathcal{O}\left(\left(\sqrt{V\log(nK)/n}\right)^{\alpha/(\alpha+1)}\right) = \epsilon + \tilde{\mathcal{O}}(n^{-\alpha/(2\alpha+2)}).$$

∎

### A.2.8 Proof of immunity results: Theorem 4 and 5

Here, we state the immunity theorems in an equivalent but different way, so that the proofs are easier to follow.

**Theorem (Immunity for one-hot vector)** *Denote $\mathcal{B} = \{e_1, \ldots, e_K\}$ to be the set of one-hot vectors.*

$$\forall \, \boldsymbol{\eta}(x) \in \mathcal{B}, \quad \arg\max \boldsymbol{\eta}(x) = \arg\max \boldsymbol{E}(x)^\top \boldsymbol{\eta}(x)$$

$$\Longleftrightarrow \quad \textit{Diagonal elements of } \boldsymbol{E}(x) \textit{ maximizes its row.}$$

*Proof.* Let $\boldsymbol{\eta}(x) = \boldsymbol{e}_y$ for some $y$, then

$$\widetilde{\boldsymbol{\eta}}(x) = \boldsymbol{E}^\top \boldsymbol{\eta}(x) = \begin{bmatrix} \mathbb{P}\left(\widetilde{Y} = 1 \mid Y = y, X = x\right) \\ \mathbb{P}\left(\widetilde{Y} = 2 \mid Y = y, X = x\right) \\ \vdots \\ \mathbb{P}\left(\widetilde{Y} = K \mid Y = y, X = x\right) \end{bmatrix} = [\boldsymbol{E}(x)]_{y,:}^\top$$

To have

$$\arg\max \widetilde{\boldsymbol{\eta}}(x) = \arg\max \, [\boldsymbol{E}(x)]_{y,:}^\top = \arg\max \boldsymbol{\eta}(x) = y$$

for any choice of $y$, it is equivalent to say that the diagonal elements of $\boldsymbol{E}(x)$ maximizes its row. ∎

**Theorem (Universal Immunity)** *Consider $K$-class classification,*

$$\forall \, \boldsymbol{\eta}(x), \quad \arg\max \boldsymbol{\eta}(x) = \arg\max \boldsymbol{E}(x)^\top \boldsymbol{\eta}(x)$$

$$\Longleftrightarrow \quad \exists \, e(x) \text{ s.t. } \forall x, e(x) \in \left[0, \frac{1}{K}\right) \text{ and}$$

$$\boldsymbol{E}(x) = \begin{bmatrix} 1 - (K-1)e(x) & e(x) & \cdots & e(x) \\ e(x) & 1 - (K-1)e(x) & \cdots & e(x) \\ \vdots & \vdots & \ddots & \vdots \\ e(x) & e(x) & \cdots & 1 - (K-1)e(x) \end{bmatrix}.$$

*Proof.*

$\Longleftarrow$: Plug $\boldsymbol{E}(x)$ into the expression

$$\widetilde{\boldsymbol{\eta}}(x) = \boldsymbol{E}^\top \boldsymbol{\eta}(x)$$

$$= \begin{bmatrix} 1 - (K-1)e(x) & e(x) & \cdots & e(x) \\ e(x) & 1 - (K-1)e(x) & \cdots & e(x) \\ \vdots & \vdots & \ddots & \vdots \\ e(x) & e(x) & \cdots & 1 - (K-1)e(x) \end{bmatrix} \boldsymbol{\eta}(x)$$

$$= \left( (1 - K \cdot e(x)) \cdot \begin{bmatrix} 1 & 0 & \cdots & 0 \\ 0 & 1 & \cdots & 0 \\ \vdots & \vdots & \ddots & \vdots \\ 0 & 0 & \cdots & 1 \end{bmatrix} + e(x) \cdot \begin{bmatrix} 1 \\ 1 \\ \vdots \\ 1 \end{bmatrix} \cdot \begin{bmatrix} 1 & \cdots & 1 \end{bmatrix} \right) \boldsymbol{\eta}(x)$$

$$= (1 - K \cdot e(x)) \, \boldsymbol{\eta}(x) + \text{ constant vector}$$

When $e(x) \in [0, \frac{1}{K})$, we have

$$\forall \boldsymbol{\eta}(x), \ \arg\max \widetilde{\boldsymbol{\eta}}(x) = \arg\max \boldsymbol{\eta}(x).$$

$\Longrightarrow$: Denote $\boldsymbol{T}(x) := \boldsymbol{E}(x)^\top$, then

$$\boldsymbol{T}(x) = \begin{bmatrix} t_{11}(x) & t_{12}(x) & \cdots & t_{1K}(x) \\ t_{21}(x) & t_{22}(x) & \cdots & t_{2K}(x) \\ \vdots & \vdots & \ddots & \vdots \\ t_{K1}(x) & t_{K2}(x) & \cdots & t_{KK}(x) \end{bmatrix}$$

has each column sum to 1. Let us consider several choices of $\boldsymbol{\eta}(x)$, which pose conditions on matrix $\boldsymbol{T}(x)$.

1) If $\boldsymbol{\eta}(x) = \begin{bmatrix} \frac{1}{K} & \frac{1}{K} & \cdots & \frac{1}{K} \end{bmatrix}^\top$, then

$$\widetilde{\boldsymbol{\eta}}(x) = \boldsymbol{T}(x)\boldsymbol{\eta}(x) = \frac{1}{K} \begin{bmatrix} t_{11}(x) + t_{12}(x) + \cdots + t_{1K}(x) \\ t_{21}(x) + t_{22}(x) + \cdots + t_{2K}(x) \\ \vdots \\ t_{K1}(x) + t_{K2}(x) + \cdots + t_{KK}(x) \end{bmatrix}.$$

To have

$$\arg\max \widetilde{\boldsymbol{\eta}}(x) = \arg\max \boldsymbol{\eta}(x) = \{1, 2, \ldots, k\},$$

all elements of $\widetilde{\boldsymbol{\eta}}(x)$ must be equal, i.e., each row of $\boldsymbol{T}(x)$ should sum to the same value. The sum of all elements in $\boldsymbol{T}(x)$ is $K$, since all column sum to 1. Therefore, each row of $\boldsymbol{T}(x)$ also sum to 1.

2) If $\boldsymbol{\eta}(x) = \begin{bmatrix} \frac{1}{K-1} & \frac{1}{K-1} & \cdots & \frac{1}{K-1} & 0 \end{bmatrix}^\top$, then

$$\widetilde{\boldsymbol{\eta}}(x) = \boldsymbol{T}(x)\boldsymbol{\eta}(x) = \frac{1}{K-1} \begin{bmatrix} t_{11}(x) + t_{12}(x) + \cdots + t_{1(K-1)}(x) \\ t_{21}(x) + t_{22}(x) + \cdots + t_{2(K-1)}(x) \\ \vdots \\ t_{(K-1)1}(x) + t_{(K-1)2}(x) + \cdots + t_{(K-1)(K-1)}(x) \\ t_{K1}(x) + t_{K2}(x) + \cdots + t_{K(K-1)}(x) \end{bmatrix}$$

$$= \frac{1}{K-1} \begin{bmatrix} 1 - t_{1K}(x) \\ 1 - t_{2K}(x) \\ \vdots \\ 1 - t_{(K-1)k}(x) \\ 1 - t_{KK}(x) \end{bmatrix}. \qquad \because \text{each row of } \boldsymbol{T}(x) \text{ sum to 1}$$

To have

$$\arg\max \widetilde{\boldsymbol{\eta}}(x) = \arg\max \boldsymbol{\eta}(x) = \{1, 2, \ldots, K-1\},$$

the first $K - 1$ elements of $\widetilde{\boldsymbol{\eta}}(x)$ must be equal (and larger than $t_{KK}(x)$), then we have

$$t_{1K}(x) = t_{2K}(x) = \cdots = t_{(K-1)K}(x).$$

In other words, all elements of the $K$-th column of $\boldsymbol{T}(x)$ are the same (except for the $(K, K)$-th element). Similarly, consider $\boldsymbol{\eta}(x)$ to be a vector that contains 0 in the $i$-th position and $\frac{1}{K-1}$ in other positions, then the general condition for $\boldsymbol{T}(x)$ is that: all elements of the $i$-th column are equal, except the $i$-th diagonal. Written explicitly,

$$\boldsymbol{T}(x) = \begin{bmatrix} t_{11}(x) & t_{12}(x) & t_{13}(x) & \cdots & t_{1K}(x) \\ t_{21}(x) & t_{22}(x) & t_{13}(x) & \cdots & t_{1K}(x) \\ t_{21}(x) & t_{12}(x) & t_{33}(x) & \cdots & t_{1K}(x) \\ \vdots & \vdots & \vdots & \ddots & \vdots \\ t_{21}(x) & t_{12}(x) & t_{13}(x) & \cdots & t_{KK}(x) \end{bmatrix}.$$

Since each row and column of $\boldsymbol{T}(x)$ sum to 1, we have

$$
\begin{cases}
0 + t_{12}(x) + t_{13}(x) + \cdots + t_{1K}(x) &= (K-1)t_{21}(x) \longleftarrow \text{ sum of first row = sum of first column} \\
t_{21}(x) + 0 + t_{13}(x) + \cdots + t_{1K}(x) &= (K-1)t_{12}(x) \\
t_{21}(x) + t_{12}(x) + 0 + \cdots + t_{1K}(x) &= (K-1)t_{13}(x) \\
&\vdots \\
t_{21}(x) + t_{12}(x) + t_{13}(x) + \cdots + 0 &= (K-1)t_{1K}(x)
\end{cases}
$$

Subtracting the first equation from the second, we have $t_{12}(x) = t_{21}(x)$. Repeating for all pairs of equations, we have $t_{21}(x) = t_{12}(x) = t_{13}(x) = \cdots = t_{1K}(x)$. What's more, all diagonal elements of $\boldsymbol{T}(x)$ will be equal. Thus,

$$
\boldsymbol{T}(x) = \begin{bmatrix}
1 - (K-1)e(x) & e(x) & \cdots & e(x) \\
e(x) & 1 - (K-1)e(x) & \cdots & e(x) \\
\vdots & \vdots & \ddots & \vdots \\
e(x) & e(x) & \cdots & 1 - (K-1)e(x)
\end{bmatrix}, \text{ where } e(x) \in [0,1].
$$

3) The final step is to determine what value $e(x)$ can take. Take $\boldsymbol{\eta}(x) = \boldsymbol{e}_y$ for some $y$, then from Theorem 4, we know that the diagonal elements of $\boldsymbol{T}(x)$ maximize their column, therefore

$$
1 - (K-1)e(x) > e(x) \quad \implies \quad e(x) \in [0, \frac{1}{K}).
$$

Finally, take any $\boldsymbol{\eta}(x)$, the $\arg\max$ is preserved by multiplying this specific choice of $\boldsymbol{T}(x)$. This concludes the $\Longleftarrow$ part. ∎

## A.3 Experimental details

### A.3.1 2D Gaussian with synthetic label noise

For 2D Gaussian mixture data, we draw from two Gaussian centered at $[1 \ 1]^\top$ and $[-1 \ -1]^\top$, with covariance matrix being identity, 200 data points from each, with label $Y = 1, 2$ respectively. To generate noisy labels, we flip every label uniformly with some probability. We use Sklearn's logistic regression (with no $\ell_2$ regularization). The experiment was conducted on AMD Ryzen 5 3600 CPU. The goal of the simulation is to experimentally verify noise immunity results in Section 6. Notice that different trial corresponds to different draw of both instances and noisy labels.

Table 2: Testing accuracy of logistic regression on gaussian mixture data with uniform label noise. "Noise rate" refers to $\mathbb{P}(\widetilde{Y} \neq Y)$, the percentage of wrong labels in the training data. As theory in Section 6 predicts, when $\mathbb{P}(\widetilde{Y} \neq Y)$ reach $50\%$, there is a sharp decrease in performance.

| Noise rate | 0 | 0.05 | 0.1 | 0.15 | 0.2 | 0.25 | 0.3 | 0.35 | 0.4 | 0.45 | 0.5 | 0.55 | 0.6 | 0.65 | 0.7 | 0.75 | 0.8 | 0.85 | 0.9 | 0.95 | 1 |
|---|---|---|---|---|---|---|---|---|---|---|---|---|---|---|---|---|---|---|---|---|---|
| Trial #1 | 93.00 | 92.83 | 92.38 | 92.08 | 91.78 | 91.93 | 92.25 | 92.90 | 91.83 | 92.58 | 74.68 | 25.12 | 9.70 | 7.73 | 7.52 | 7.25 | 7.38 | 7.15 | 7.18 | 7.10 | 7.00 |
| Trial #2 | 91.73 | 91.60 | 92.05 | 91.63 | 91.78 | 91.78 | 91.68 | 91.63 | 91.55 | 91.48 | 80.40 | 21.10 | 9.93 | 8.55 | 8.38 | 8.22 | 8.20 | 8.35 | 8.33 | 8.40 | 8.28 |
| Trial #3 | 92.73 | 92.75 | 92.78 | 92.78 | 92.58 | 92.45 | 91.68 | 88.15 | 82.58 | 59.83 | 49.53 | 35.80 | 21.28 | 14.35 | 9.33 | 8.53 | 8.12 | 7.70 | 7.13 | 7.23 | 7.28 |
| Trial #4 | 91.55 | 91.58 | 91.60 | 91.63 | 91.68 | 91.60 | 91.25 | 90.98 | 89.98 | 86.38 | 60.53 | 9.95 | 8.75 | 10.00 | 10.45 | 9.08 | 9.00 | 9.53 | 9.20 | 9.03 | 8.45 |
| Trial #5 | 91.55 | 91.58 | 91.60 | 91.63 | 91.68 | 91.60 | 91.25 | 90.98 | 89.98 | 86.38 | 60.53 | 9.95 | 8.75 | 10.00 | 10.45 | 9.08 | 9.00 | 9.53 | 9.20 | 9.03 | 8.45 |
| Mean | 92.11 | 92.07 | 92.08 | 91.95 | 91.90 | 91.87 | 91.62 | 90.93 | 89.18 | 83.33 | 65.13 | 20.40 | 11.68 | 10.10 | 9.23 | 8.43 | 8.34 | 8.45 | 8.21 | 8.16 | 7.89 |
| Std | 0.70 | 0.66 | 0.51 | 0.50 | 0.38 | 0.35 | 0.41 | 1.74 | 3.79 | 13.44 | 12.35 | 10.94 | 5.39 | 2.56 | 1.29 | 0.75 | 0.68 | 1.07 | 1.03 | 0.94 | 0.70 |

### A.3.2 MNIST with synthetic label noise

We flip the clean training label of MNIST (http://yann.lecun.com/exdb/mnist/) uniformly (to any of the wrong classes). We use a shallow neural network with two convolution layers and two fully connected layers. We train with stochastic gradient descent with learning rate $0.01$ for $10$ epochs, batch size equals $64$. We use the same hyperparamters for all tests. The experiments were conducted on a single NVIDIA GTX 1660S GPU. The goal of the simulation is to experimentally verify noise immunity results in Section 6. Here randomness corresponds to different realization of noisy labels and stochastic gradient descent.

Table 3: Testing accuracy of a shallow CNN (2 conv layers with 2 fully connected layers) on MNIST with uniform label noise. "Noise rate" refers to $\mathbb{P}\big(\widetilde{Y} \neq Y\big)$, the percentage of wrong labels in the training data. As theory in Section 6 predicts, when $\mathbb{P}\big(\widetilde{Y} \neq Y\big)$ reach 90%, there is a sharp decrease in performance.

| Noise rate | 0 | 0.05 | 0.1 | 0.15 | 0.2 | 0.25 | 0.3 | 0.35 | 0.4 | 0.45 | 0.5 | 0.55 | 0.6 | 0.65 | 0.7 | 0.75 | 0.8 | 0.85 | 0.9 | 0.95 | 1 |
|---|---|---|---|---|---|---|---|---|---|---|---|---|---|---|---|---|---|---|---|---|---|
| Trial #1 | 98.97 | 98.89 | 98.81 | 98.46 | 98.49 | 98.16 | 98.46 | 98.07 | 97.98 | 97.57 | 97.88 | 97.84 | 97.19 | 97.10 | 96.70 | 95.02 | 89.00 | 83.72 | 11.58 | 0.17 | 0.03 |
| Trial #2 | 98.88 | 98.73 | 98.94 | 98.55 | 98.72 | 98.66 | 98.50 | 98.24 | 98.15 | 98.23 | 97.86 | 97.98 | 97.70 | 97.10 | 96.91 | 95.76 | 91.99 | 88.49 | 9.99 | 0.08 | 0.04 |
| Trial #3 | 99.00 | 99.04 | 98.86 | 98.56 | 98.69 | 98.66 | 98.51 | 98.49 | 98.37 | 98.25 | 98.25 | 97.39 | 97.37 | 97.18 | 96.66 | 94.88 | 92.15 | 81.48 | 6.19 | 0.14 | 0.04 |
| Trial #4 | 99.04 | 98.86 | 98.70 | 98.76 | 98.83 | 98.65 | 98.34 | 98.42 | 98.58 | 98.47 | 98.00 | 97.41 | 97.63 | 97.09 | 96.46 | 95.94 | 93.19 | 84.78 | 8.68 | 0.19 | 0.01 |
| Trial #5 | 99.05 | 98.58 | 98.89 | 98.82 | 98.72 | 98.83 | 98.34 | 98.55 | 98.40 | 98.38 | 98.01 | 97.31 | 97.33 | 96.21 | 96.29 | 94.92 | 90.38 | 85.84 | 8.98 | 0.13 | 0.08 |
| Mean | 98.99 | 98.82 | 98.84 | 98.63 | 98.69 | 98.59 | 98.43 | 98.35 | 98.30 | 98.18 | 98.00 | 97.59 | 97.44 | 96.94 | 96.60 | 95.30 | 91.34 | 84.86 | 9.08 | 0.14 | 0.04 |
| Std | 0.07 | 0.17 | 0.09 | 0.15 | 0.12 | 0.25 | 0.08 | 0.20 | 0.23 | 0.36 | 0.16 | 0.30 | 0.21 | 0.41 | 0.24 | 0.51 | 1.65 | 2.59 | 1.98 | 0.04 | 0.03 |

## A.3.3 CIFAR with synthetic label noise

We flip the clean training label of CIFAR-10 (https://www.cs.toronto.edu/~kriz/cifar.html) uniformly (to any of the wrong classes). To have a fair comparison between different methods, we fix the realization of noisy labels. Follow the 2-step procedure described in Section 7, we use different pre-trained neural networks as feature extractor: forward-passing the training image through the network and record the feature. Then use sklearn's (https://scikit-learn.org/stable/) logistic regression function to fit the (feature, noisy label) pair in a full batch manner. We pre-specify a range of values for $\ell_2$ regularization ($\{0.0001, 0.001, 0.01, 0.1, 1, 10, 100\}$ ) and number of iterations for lbfgs optimizer ($\{10, 20, 50, 100\}$), then do cross-validation on noisy data to pick the best hyper-parameters. We use the same range of hyper-parameters in all tests. The experiments were conducted on a single NVIDIA Tesla V100 GPU. The result is deterministic.

Table 4: Peformance on CIFAR-10 with synthetic label noise. We apply linear model on top of different feature extractors: "ResNet-50 TL" refers to using a pre-trained ResNet-50 on ImageNet [Deng et al., 2009] (available in Pytorch model library) in a transfer learning fashion, "ResNet-50 SSL" refers to using a pre-trained ResNet-50 on unlabeled CIFAR data with self-supervised loss [Chen et al., 2020] (publicly downloadable weights https://github.com/ContrastToDivide/C2D?tab=readme-ov-file) and "DINOv2 SSL" refers to using the self-supervised foundation model DINOv2 [Oquab et al., 2023] (available at https://github.com/facebookresearch/dinov2) as the feature extractor. "Noise rate" refers to $\mathbb{P}\big(\widetilde{Y} \neq Y\big)$, the percentage of wrong labels in the training data. As theory in Section 6 predicts, when $\mathbb{P}\big(\widetilde{Y} \neq Y\big)$ reach 90%, there is a sharp decrease in performance. We employed Python's sklearn logistic regression and cross-validation functions without data augmentation. The results are deterministic.

| Noise rate | 0 | 0.1 | 0.2 | 0.3 | 0.4 | 0.5 | 0.6 | 0.7 | 0.8 | 0.85 | 0.9 | 0.95 | 1 |
|---|---|---|---|---|---|---|---|---|---|---|---|---|---|
| Linear | 41.37 | 41.09 | 40.97 | 40.37 | 40.45 | 39.44 | 37.28 | 35.20 | 26.74 | 18.00 | 10.28 | 5.50 | 3.92 |
| Linear + ResNet-50 TL | 90.17 | 89.58 | 89.01 | 88.27 | 87.55 | 87.28 | 86.40 | 85.01 | 82.03 | 74.02 | 10.82 | 1.47 | 0.26 |
| Linear + ResNet-50 SSL | 92.48 | 92.26 | 91.74 | 91.46 | 91.13 | 90.33 | 91.07 | 90.99 | 89.11 | 83.89 | 10.08 | 1.31 | 0.34 |
| Linear + DINOv2 SSL | 99.25 | 99.27 | 99.23 | 99.14 | 99.10 | 99.11 | 99.02 | 98.84 | 95.50 | 76.91 | 10.13 | 0.92 | 0.03 |

## A.3.4 CIFAR with human label error

We load the noisy human labels provided by http://noisylabels.com/, then follow exact the same procedure as above.

Table 5: Performance on CIFAR-N dataset (`http://noisylabels.com/`) in terms of testing accuracy. "Aggre", "Rand1", ..., "Noisy" denote various types of human label noise. We apply linear model on top of different feature extractors: "ResNet-50 TL" refers to using a pre-trained ResNet-50 on ImageNet [Deng et al., 2009] in a transfer learning fashion, "ResNet-50 SSL" refers to using a pre-trained ResNet-50 on unlabeled CIFAR data with self-supervised loss [Chen et al., 2020] and "DINOv2 SSL" refers to using the self-supervised foundation model DINOv2 [Oquab et al., 2023] as the feature extractor. We employed Python's sklearn logistic regression and cross-validation functions without data augmentation; the results are deterministic and directly reproducible.

| Methods | CIFAR-10N | | | | | CIFAR-100N |
|---|---|---|---|---|---|---|
| | Aggre | Rand1 | Rand2 | Rand3 | Worst | Noisy |
| Linear | 40.73 | 40.41 | 40.31 | 40.63 | 38.43 | 16.61 |
| Linear + ResNet-50 TL | 89.18 | 88.63 | 88.61 | 88.66 | 85.32 | 62.89 |
| Linear + ResNet-50 SSL | 91.78 | 91.66 | 91.39 | 91.28 | 87.84 | 57.95 |
| Linear + DINOv2 SSL | 98.69 | 98.80 | 98.65 | 98.67 | 95.71 | 83.17 |

## A.4 Additional experiments

### A.4.1 Linear probing, then fine tuning (LP-FT)

We study whether 'linear probing, then fine tuning' (LP-FT) [Kumar et al., 2022] works better than linear probing (LP) only, in label noise learning scenario.

Table 6: Performance on CIFAR-N dataset (`http://noisylabels.com/`) in terms of testing accuracy. "Clean" refers to no label noise, "Aggre", "Rand1", ..., "Noisy" denote various types of human label noise. We compare the testing accuracy of LP-FT versus LP only, over different feature extractors: "ResNet-50 TL" refers to using a pre-trained ResNet-50 on ImageNet [Deng et al., 2009] in a transfer learning fashion, "ResNet-50 SSL" refers to using a pre-trained ResNet-50 on unlabeled CIFAR data with contrastive loss [Chen et al., 2020] and "DINOv2 (small) SSL" refers to using a light version of the self-supervised foundation model DINOv2 [Oquab et al., 2023] as the feature extractor.

| Feature | Method | CIFAR-10N | | | | | | CIFAR-100N | |
|---|---|---|---|---|---|---|---|---|---|
| | | Clean | Aggre | Rand1 | Rand2 | Rand3 | Worst | Clean | Noisy |
| ResNet-50 TL | LP (ours) | 90.17 | 89.18 | **88.63** | 88.61 | **88.66** | **85.32** | 71.79 | 62.89 |
| | LP-FT | **95.94** | **92.03** | 88.55 | 87.78 | 87.82 | 71.88 | **82.3** | **63.85** |
| ResNet-50 SSL | LP (ours) | 92.54 | **91.78** | **91.66** | **91.46** | **91.17** | **87.85** | 69.88 | **57.98** |
| | LP-FT | **94.11** | 89.11 | 84.49 | 83.75 | 84.15 | 65.00 | **74.41** | 54.49 |
| DINOv2 (small) SSL | LP (ours) | 96.09 | **94.8** | **94.39** | **94.42** | **94.35** | **91.14** | 83.82 | **72.46** |
| | LP-FT | **98.23** | 93.29 | 88.03 | 87.27 | 86.94 | 67.42 | **89.97** | 64.81 |

### A.4.2 Robust learning strategy over DINOv2 feature

This section examines how different robust learning strategy works over DINOv2 feature, compared with only training with cross entropy.

Table 7: Comparison of different noise robust methods on DINOv2 features. Training a linear classifier with cross entropy (CE) loss is the baseline. We compare it with robust losses: mean absolute error (MAE) loss [Ghosh and Kumar, 2017, Ma and Fattahi, 2022], sigmoid loss [Ghosh et al., 2015], and regularized approaches: 'Early-Learning Regularization' (ELR) [Liu et al., 2020], 'Sharpness Aware Minimization' (SAM) [Foret et al., 2021].

| Feature | Method | CIFAR-10N | | | | | | CIFAR-100N | |
|---|---|---|---|---|---|---|---|---|---|
| | | Clean | Aggre | Rand1 | Rand2 | Rand3 | Worst | Clean | Noisy |
| DINOv2 SSL | CE | 99.25 | 98.69 | 98.8 | 98.65 | 98.67 | 95.71 | **92.85** | **83.17** |
| | MAE | **99.27** | **99.04** | **99.01** | **99.09** | **99.11** | 95.55 | 90.68 | 82.55 |
| | Sigmoid | 99.26 | 98.86 | 98.91 | 98.87 | 98.96 | **96.66** | 92.82 | 82.03 |
| | ELR | 99.09 | 98.49 | 98.62 | 98.53 | 98.56 | 95.60 | 89.99 | 82.75 |
| | SAM | 99.09 | 97.66 | 98.47 | 98.53 | 98.47 | 95.47 | 89.97 | 82.85 |

### A.4.3 Synthetic instance-dependent label noise

Table 8: We synthetically corrupt labels of CIFAR-10 according to Xia et al. [2020], and compare our NI-ERM with the 'Part-dependent matrix estimation' (PTD) method proposed in that same paper.

| Method \ Noise rate | 10 % | 20 % | 30 % | 40 % | 50 % |
|---|---|---|---|---|---|
| PTD | 79.01 | 76.05 | 72.28 | 58.62 | 53.98 |
| NI-ERM (ours) | **99.11** | **98.94** | **98.20** | **93.35** | **74.67** |

