# OpenReview forum: "Label Noise: Ignorance Is Bliss"
_NeurIPS.cc/2024/Conference — NeurIPS 2024 poster_

### Official Review · Reviewer_G7Jo · 2024-07-09

**Soundness:** 3
**Presentation:** 4
**Contribution:** 3
**Rating:** 7
**Confidence:** 3

**Summary:**

This paper presents a theoretical framework for learning under multi-class, instance-dependent label noise. It introduces the novel concept of Relative Signal Strength (RSS) to measure the impact of noise and uses it to derive upper and lower bounds on excess risk. Notably, it proves the minimax optimality of Noise Ignorant Empirical Risk Minimization (NI-ERM) and provides conditions for immunity to label noise. Bridging theory and practice, the paper proposes a two-step 'feature extraction + NI-ERM' approach, achieving state-of-the-art performance on the CIFAR-N dataset.

**Strengths:**

1. Theoretical Depth:
This paper introduces the novel concept of Relative Signal Strength (RSS) to quantify the impact of noise in label learning. Using RSS, the authors mathematically derive precise upper and lower bounds for excess risk.

2. Combination of Theory and Practicality:
The research seamlessly bridges theory and practice by developing a two-step 'feature extraction + NI-ERM' approach based on their theoretical analysis. This method achieves high performance using simple linear classifiers instead of complex deep learning models, validating the theoretical predictions about the optimality of NI-ERM in practical applications. This successful translation of theoretical insights into effective practical methods is a significant strength of the paper.

3. Surprising Results:
One of the most striking aspects of this work is the mathematical proof that simple Noise Ignorant Empirical Risk Minimization (NI-ERM) is minimax optimal. The authors theoretically guarantee accurate learning even under high noise levels, such as up to 90% in 10-class problems. These surprising findings challenge conventional wisdom about the necessity of complex noise handling techniques and achieve state-of-the-art performance with a remarkably simple approach.

4. Experimental Validation:
This paper provides experimental validation of its theoretical claims by achieving top performance on real-world noisy datasets like CIFAR-10N and CIFAR-100N. They demonstrate that performance changes with increasing noise levels align closely with their theoretical predictions, providing empirical support for their theoretical framework.

5. Practicality:
The proposed method achieves high performance without relying on data augmentation or complex hyperparameter tuning

**Weaknesses:**

1. Insufficient Introduction:
The introduction lacks a comprehensive overview of the paper's content and contributions. A more detailed exposition of the overall approach and key findings would better prepare readers and emphasize the paper's significance. This could include a clearer roadmap of the theoretical and practical aspects of the work.

2. Validity of Relative Signal Strength (RSS) Definition:
The definition of RSS is not intuitive and lacks a sufficient explanation of how it represents the signal content of the noisy distribution relative to the clean distribution. While examples are provided, they primarily demonstrate RSS calculation rather than explaining the fundamental reasoning behind its definition. Since all theoretical results in the paper are based on the RSS definition, it is crucial to establish whether RSS accurately represents the degree to which the noisy distribution contains signal compared to the clean distribution. A more thorough justification and explanation would strengthen the paper's theoretical foundation. I will raise a question about the definition of RSS in the question section to seek further clarification on this point.

3. Experimental Validity Concerns:
The two-step method employs pre-trained feature extractors. While this demonstrates the effectiveness of NI-ERM, it may not provide a fair comparison with other methods. To more convincingly demonstrate the superiority of NI-ERM, the paper could:
a) Compare NI-ERM using feature extractors from models trained with other SOTA noisy label methods.
b) Apply SOTA methods to the same high-quality feature extractors used for NI-ERM, training only the final linear layer for a fairer comparison.
These additional experiments would provide a more robust validation of the NI-ERM approach in practical scenarios.

**Questions:**

1. Definition of RSS:
I am curious about how RSS measures certainty about the label. The definition of RSS does not seem intuitive. Generally, in deep learning, KL divergence is used to measure the difference between two probability distributions. The definition of RSS shows a different tendency compared to KL divergence. For example, let's consider three probability vectors: p1 = [0.05, 0.7, 0.25], p2 = [0.25, 0.7, 0.05], and p3 = [0.1, 0.6, 0.3]. The RSS between p1 and p2 is 1.44, and the RSS between p1 and p3 is 0.67. If we assume p1 is the clean distribution, it can be interpreted that p2 contains more label information. However, the KL divergence between p1 and p2 is 1.96, and between p1 and p3 is 1.05, indicating that p1 and p3 are closer in terms of distribution. Is there a justification for claiming that p2 contains more label information despite having a higher KL divergence?

2. Gap between Theory and Practice:
While the paper bridges the gap between theory and experiments by using pretrained feature extractors, as mentioned in line 254, generally training the feature extractor with NI-ERM does not perform well. This differs from the theory presented in the paper. What do you think is the reason for this discrepancy?

**Limitations:**

Limited Application to Classification Problems:
While noisy label problems exist in various tasks, the theoretical analysis and experimental validation in this paper focus solely on classification problems, particularly image classification tasks.

---

> ### Author Rebuttal · Authors · 2024-08-05
>
> Thank you for taking the time to review our paper. The "strength" section of your review indeed summarizes our contributions. We also especially thank you for bringing up the point about the relation of RSS and KL divergence. In the next version, we will include the example you provided and explain the relations between the two concepts.
>
> ## Regarding "Weaknesses":
>
> > 1. Insufficient Introduction: ... A more detailed exposition of the overall approach and key findings would better prepare readers and emphasize the paper's significance.
>
> Thank you for this suggestion. We will certainly include a paragraph outlining the content and contributions in the next version.
>
> > 2. Validity of Relative Signal Strength (RSS) Definition: The definition of RSS is **not intuitive and lacks a sufficient explanation** of how it represents the signal content of the noisy distribution relative to the clean distribution. While examples are provided, they primarily demonstrate RSS calculation rather than explaining the fundamental reasoning behind its definition.
>
> Thank you again for this comment. Fortunately, there are a few lines from the appendix (line 526-528) that we will move to the main text that show precisely where the definition of RSS comes in to play.
>
> In short, the clean excess risk is
> $$
> R(f) - R^* = \int (\max_i [\eta(x)]_i - [\eta(x)]_f ) d P_X(x),
> $$
> where we see the denominator of RSS.
> While the noisy excess risk is
> $$
> \widetilde{R}(f) - \widetilde{R}^* = \int (\max_i [\widetilde{\eta}(x)]_i - [\widetilde{\eta}(x)]_f ) d P_X(x),
> $$
> where we see the numerator of RSS.
>
> RSS is the right definition because it is what **pops up naturally** when bounding the clean excess risk with the noisy one.
>
> As for more intuition, see also our response to your first question, below.
>
> > Experimental Validity Concerns... To more convincingly demonstrate the superiority of NI-ERM, the paper could: a) Compare NI-ERM using feature extractors from models trained with other SOTA noisy label methods. b) Apply SOTA methods to the same high-quality feature extractors used for NI-ERM, training only the final linear layer for a fairer comparison.
>
> These are reasonable requests. For:
>
> a) Although we have not had time to run these experiments, we already have some insight into what would happen: the performance relates to the "feature quality", assessed in terms of classification accuracy with no noise, see line 279-282.
>
> b) We have ran additional experiments by fixing the "high-quality feature" and training the final linear layer with different robust losses [2-3] and robust training procedures [4], see Table 2 in the pdf attached to the global rebuttal. Our NI-ERM approach is highly competitive.
>
>
> ## Regarding "Questions":
>
> > 1.	Definition of RSS: I am curious about how RSS measures certainty about the label... The definition of RSS shows a different tendency compared to **KL divergence**. For example, let's consider three probability vectors: p1 = [0.05, 0.7, 0.25], p2 = [0.25, 0.7, 0.05], and p3 = [0.1, 0.6, 0.3]. The RSS between p1 and p2 is 1.44, and the RSS between p1 and p3 is 0.67. If we assume p1 is the clean distribution, it can be interpreted that p2 contains more label information. However, the KL divergence between p1 and p2 is 1.96, and between p1 and p3 is 1.05, indicating that p1 and p3 are closer in terms of distribution. Is there a justification for claiming that p2 contains more label information despite having a higher KL divergence?
>
> Thanks for bring up this excellent point.
> The short answer is:
> **KL divergence considers the similarity between two (whole) distributions, while the task of classification only focuses on predicting the $\arg \max$.**
>
> In the example you mentioned, $p_1$ is the clean class distribution, and $p_2, p_3$ can be viewed as two noisy copies of it. $p_3$ is closer to $p_1$ in terms of KL divergence, but $p_2$ provides more information in terms of predicting the $\arg \max$ of $p_1$. There is no conflict, intuitively: the difference of $p_1$ and $p_2$ lies in that the probability of being class 1 and 3 got swapped, but the "margin", aka gap between the largest probability and second largest, is still $0.7 - 0.25 = 0.45$. While in $p_3$, the "margin" is $0.6 - 0.3 = 0.3$, which is smaller than $p_2$.
>
> To conclude, although $p_1$ and $p_3$ are "closer" in terms of distribution, $p_2$ provides more information regarding predicting the $\arg \max$ of $p_1$ than $p_3$ does.
>
> We would like to include your example in our paper and write a paragraph about the relation between KL divergence and RSS.
>
> > 2. Gap between Theory and Practice: While the paper bridges the gap between theory and experiments by using pretrained feature extractors, as mentioned in line 254, generally training the feature extractor with NI-ERM does not perform well. This differs from the theory presented in the paper. What do you think is the reason for this discrepancy?
>
> This is because the Natarajan dimension of a large neural net is too big, and therefore the upper bound in Thm 2 is vacuous (bigger than 1). With a lot more data, the bound would be meaningful and NI-ERM would probably work well for end-to-end training (although the amount of data and computational resources required would probably be prohibitive). For a linear classifier, the Natarajan dimension is upper bounded by the dimension of the self-supervised features ([1], Thm 29.7), thus providing much better control on the excess risk.
>
> Reference:
>
> [1] Shai Shalev-Shwartz and Shai Ben-David. Understanding machine learning: From theory to algorithms, 2014.
>
> [2] Aritra Ghosh and Himanshu Kumar. Robust loss functions under label noise for deep neural networks. AAAI, 2017.
>
> [3] Aritra Ghosh, et al. Making risk minimization tolerant to label noise. Neurocomputing, 2015
>
> [4] Pierre Foret, et al. Sharpness-aware minimization for efficiently improving generalization. ICLR, 2021.

---

> ### Comment · Reviewer_G7Jo · 2024-08-11
> **Main Concerns Addressed by Authors' Response**
>
> Thank you for the author's response and new experiments. The author's explanation has alleviated many of my concerns. I now understand why the authors set the RSS in that way and how you tried to capture the signal of the clean posterior differently from KL divergence. While I'm still not fully convinced that RSS is the best definition for representing how much signal from the clean distribution is included, I believe it is sufficient for the proposed theories to have meaningful implications. I will raise my score to accept.

---

> > ### Author Response · Authors · 2024-08-11
> >
> > Thank you for your response.
> > Your review is very beneficial and definitely helps the paper to be better framed.
> > We really appreciate that.
> > Thanks.

---

### Official Review · Reviewer_1ugH · 2024-07-13

**Soundness:** 3
**Presentation:** 3
**Contribution:** 3
**Rating:** 8
**Confidence:** 3

**Summary:**

The work provides a new insight on how to deal with instance-dependent label noise under the context of multi-class classification problem. Under certain conditions, they prove that training a classifier as if there is no noisy labels is the best course of action.
This idea is presented in details supported by several theorems with finite sample analysis as well as by several experiments on synthetic and a real dataset (CIFAR-10N).

**Strengths:**

I enjoy reading this paper. The work presents a surprisingly simple idea to deal with noisy labels, especially instance-dependent noise, and show that it is the optimal learning strategy under some conditions.
- The authors introduce the Relative Signal Strength (RSS) to quantify how much noisy label distribution can reveal about the clean label distribution.
- Based on this measure, the authors suggest a class of noisy label learning problems that can be guaranteed to be solvable (Theorem 2) by just ignoring the existence of noisy labels. This is somewhat still intuitive because RSS is positive and the excessive risk is measured based on 0-1 loss.
- They also present a min-max analysis to show that for the same class of problems, the above learning strategy achieves an optimal rate. This is really a surprising and intriguing result.
- Lastly, they show how applicable this class of problems can be in practice (Theorem 4, 5): the label noise probability vector should have the same argmax as the clean probability vector.

**Weaknesses:**

- The key condition for ignoring the existence of noisy labels to work is to have $\mathcal{A}_0 = \mathcal{X}$ (or $\mathcal{A}_0$ covers most of $\mathcal{X}$), which in turns requires $\text{argmax } \widetilde{\eta}(x)  = \text{argmax } \eta(x)$. This condition would be violated if certain classes are tricky and easy to mistaken one as another, for example, labeling leopard, lion,  cheetah, tiger. Therefore, the condition that \kappa > 0 might be restrictive in this regard.
On the other hand, it would be intuitively feasible to still learn a classifier under this type of noise as long as one mistakes a class with another class in a consistent way. In that case, having permutation ambiguity is inevitable, but it is not a very detrimental as the cost of post-processing to disambiguating permutation is relatively cheap.
- The experiment section is limited:
   + How is performance of the proposal compared to baselines under various noise level?
   + More baselines should be included, such as BLTM[1], MEIDTM[2], MaxMIG[3].

[1] Yang, Shuo, et al. "Estimating instance-dependent bayes-label transition matrix using a deep neural network." International Conference on Machine Learning. PMLR, 2022.

[2] Cheng, De, et al. "Instance-dependent label-noise learning with manifold-regularized transition matrix estimation." Proceedings of the IEEE/CVF Conference on Computer Vision and Pattern Recognition. 2022.

[3] Cao, Peng, et al. "Max-mig: an information theoretic approach for joint learning from crowds." arXiv preprint arXiv:1905.13436 (2019).

**Questions:**

- Does the analysis depend on any particular loss used during training?

**Limitations:**

yes

---

> ### Author Rebuttal · Authors · 2024-08-05
>
> Thank you for taking the time to review our paper. We are especially glad that you enjoyed reading our paper.
>
> > The key condition for ignoring the existence of noisy labels to work is to have A0=X (or A0 covers most of X), which in turns requires $\arg \max \widetilde{\eta}(x)= \arg \max \eta(x)$. This condition would be violated if certain classes are tricky and easy to mistaken one as another, for example, labeling leopard, lion, cheetah, tiger. Therefore, the condition that $\kappa > 0$ might be restrictive in this regard. On the other hand, it would be intuitively feasible to still learn a classifier under this type of noise as long as one mistakes a class with another class in a consistent way. In that case, having permutation ambiguity is inevitable, but it is not a very detrimental as the cost of post-processing to disambiguating permutation is relatively cheap.
>
> That is an interesting scenario. It seems to us that "disambiguate a label permutation" requires additional information (e.g., human feedback), which is not a part of our problem statement.
>
> If we allow further "post-processing", that would be an interesting research problem worth exploring.
>
>
> > Does the analysis depend on any particular loss used during training?
>
> The theoretical analysis is on zero-one loss. To incorporate surrogate losses, one can use the classification-calibration [1] argument.
>
> As for practical performance, we have tried more than 10 different multi-class losses, they end up performing similarly. We could incorporate that in our next version.
>
> As for now, Table 2 in the attached pdf file to the global rebuttal shows the comparison of cross entropy to several "noise robust losses" [2-3], their results are comparable.
>
>
> > The experiment section is limited... More baselines should be included, such as BLTM, MEIDTM, MaxMIG.
>
> Thanks for mentioning these papers, we will include them in our next version.
>
> At this point, due to time constraint, the additional experiments we have ran are shown in the attached pdf to the global rebuttal. Feel free to take a look, thanks.
>
>
> Reference:
>
> [1] Peter Bartlett, Michael Jordan, and Jon McAuliffe. "Convexity, classification, and risk bounds." Journal of the American Statistical Association, 2006.
>
> [2] Aritra Ghosh and Himanshu Kumar. Robust loss functions under label noise for deep neural networks. In Proceedings of the AAAI conference on artificial intelligence, 2017.
>
> [3] Aritra Ghosh, et al. Making risk minimization tolerant to label noise. Neurocomputing, 2015

---

> > ### Comment · Reviewer_1ugH · 2024-08-12
> >
> > Dear authors,
> >
> > Thank you for addressing my comments. All my questions are answered and I have no further comments. I will keep my initial rating.

---

### Official Review · Reviewer_V7Mw · 2024-07-15

**Soundness:** 3
**Presentation:** 4
**Contribution:** 3
**Rating:** 5
**Confidence:** 2

**Summary:**

The author proposes a Label Noise Learning (LNL) method that assumes a noise transition matrix. The author introduces the concept of Relative Signal Strength (RSS), which is calculated as the ratio of the signal difference between the true prediction and the prediction under label noise. The author demonstrates that the set satisfying the condition of RSS being greater than zero is noise immune. Additionally, the author provides a method to define a noise transition matrix that upholds this condition.

**Strengths:**

The author presents a novel perspective for solving the LNL problem. The interpretations of RSS proposed by the author are sufficiently insightful for peers in the LNL field. Moreover, the author's method is well-suited for use with pretrained feature extractors and aligns with current trends in the AI field.

**Weaknesses:**

There are two major concerns. The first is the low reproducibility. Despite the robust interpretation of the proposed method, the lack of a specific algorithm makes it challenging to conceptualize a clear learning approach. Specifically, it would be beneficial to provide examples of e(x) in Theorem 5, as well as pseudocode for feature extraction and the derivation of the transition matrix. Secondly, it is challenging to interpret the advantages of the proposed method from the experiments. (1) The author conducted experiments solely on light synthetic data, making it difficult to ascertain the effectiveness of the theoretically-based method in practical noise scenarios. (2) The author employs a highly trained pretrained model, which introduces an unfair factor in comparisons with other methods. Notably, there is experimental evidence suggesting that self-supervised pre-training can enhance the performance of existing methods (https://arxiv.org/pdf/2103.13646v2). I recommend including the author's method applied to various pretrained models in Table 1, or adding the use of pretrained models for existing methods to provide a fair comparison.

**Questions:**

I have given high marks to the author's novel perspective and the reasonable interpretations and proof methods presented. Despite the proposed method's low reproducibility and unproven practical performance, it has the potential to positively influence other research in the LNL field. Therefore, I have assigned a rating of "weak accept". If the author provides more detailed treatment of the proposed method and richer empirical interpretations, I believe the paper would be strong enough for acceptance.

**Limitations:**

-

---

> ### Author Rebuttal · Authors · 2024-08-05
>
> Thank you for taking the time to review our paper.
>
> > I have given high marks to the author's novel perspective and the reasonable interpretations and proof methods presented... Therefore, I have assigned a rating of "weak accept". If the author provides more detailed treatment of the proposed method and richer empirical interpretations, I believe the paper would be strong enough for acceptance.
>
> We are glad that you find our paper helpful. **A small point: the current rating seems to corresponding to "borderline accept" rather than "weak accept".** Thanks.
>
> >There are two major concerns. The first is the low reproducibility. Despite the robust interpretation of the proposed method, the lack of a specific algorithm makes it challenging to conceptualize a clear learning approach. Specifically, it would be beneficial to provide examples of e(x) in Theorem 5,
>
> Actually, $e(x)$ is not part of our algorithm. It is a quantity that describes a theoretical condition under which NI-ERM is consistent. It is not needed by the algorithm.
>
> > it would be beneficial to provide ... pseudocode for feature extraction and the derivation of the transition matrix.
>
> Actually, we do not propose a method for feature extraction. We simply use existing methods (e.g., pre-trained ResNet available in pytorch model zoo).
>
> We also do not estimate a noise transition matrix. This is a mathematical object that we use to state theorems about the performance of NI-ERM, but it is not a quantity that is needed as input to an algorithm.
>
> We have a description of our practical method, which is described in steps 1 and 2 in Section 6 (line 260-263). It is actually a meta-algorithm, allowing the user flexibility in choosing the feature extractor and empirical risk minimizer.
>
> > Secondly, it is challenging to interpret the advantages of the proposed method from the experiments. (1) The author conducted experiments solely on light synthetic data, making it difficult to ascertain the effectiveness of the theoretically-based method in practical noise scenarios.
>
> In paper, we have results for real data (MNIST, CIFAR), including the noisy CIFAR dataset which has real-world (human generated) noisy labels.
>
> In response to your (and other reviewers') requests, we have also performed some additional experiments. See the results in the pdf file attached to global response (and our responses to other reviewers), those demonstrates our NI-ERM is hard to beat.
>
> > (2) The author employs a highly trained pretrained model, which introduces an unfair factor in comparisons with other methods. Notably, there is experimental evidence suggesting that self-supervised pre-training can enhance the performance of existing methods (https://arxiv.org/pdf/2103.13646v2). I recommend including the author's method applied to various pretrained models in Table 1, or adding the use of pretrained models for existing methods to provide a fair comparison.
>
> Thanks for bringing this up, we will include the paper in the reference.
>
> In respond to your request, we have ran additional experiments, all based on the same pretrained model, see Table 2 in the pdf attached to the global response.
>
>
> Notice:
>
> the "noise rate of $90\\%$" in Table 1 of the referred paper (https://arxiv.org/pdf/2103.13646v2) corresponds to "actual noise rate $P(Y \neq \widetilde{Y}) = 0.90 \times (1-1/10) = 81 \\%$".

---

> > ### Comment · Reviewer_V7Mw · 2024-08-12
> >
> > Thank you for the detailed review. I realized that I had misunderstood the paper, and through the rebuttal and the opinions of other reviewers, I now understand that the author was justifying the use of the linear probing approach. Thanks to this, my concerns about low reproducibility have been alleviated, but I still have other remaining concerns. As highlighted in the global rebuttal, the power of a well-trained feature extractor cannot be overlooked. I have an additional question at this point. Did the other methods also involve training only the classifier? I am curious whether the author’s theoretical interpretation is empirically valid: if the performance of other methods remains high with the well-trained feature extractor being frozen, the practical significance of the author’s findings would be greatly diminished.

---

> > > ### Comment · Reviewer_V7Mw · 2024-08-12
> > >
> > > I have lowered my confidence level due to the realization of my misunderstanding.

---

> > > ### Author Response · Authors · 2024-08-13
> > >
> > > Thank you for your reply.
> > >
> > > > I now understand that the author was justifying the use of the linear probing approach.
> > >
> > > Practically, yes. Motivated by the theory, we are demonstrating (Noise-Ignorant) linear probing itself as a promising approach for label noise.
> > >
> > > > Did the other methods also involve training only the classifier? I am curious whether the author’s theoretical interpretation is empirically valid: if the performance of other methods remains high with the well-trained feature extractor being frozen, the practical significance of the author’s findings would be greatly diminished.
> > >
> > > As for Table 2 in the attached pdf file to the global rebuttal, yes, the feature extractor is frozen.
> > >
> > > The simple method performs as good as (and sometimes better than) other more complicated "robust" methods, which shall be good news for practitioners.
> > >
> > > Thanks.

---

> ### Comment · Reviewer_V7Mw · 2024-08-13
>
> Thank you for your reply. As author continues to assert, one of the undeniable contributions of the author is demonstrating that LP is a simple yet competitive method using the RSS metric. However, the limited applicability of the proposed method may provide little assistance to colleagues researching LNL. Therefore, my concern has been partially addressed, and I will maintain my original evaluation: borderline accept.

---

### Official Review · Reviewer_JX6n · 2024-07-18

**Soundness:** 3
**Presentation:** 3
**Contribution:** 3
**Rating:** 6
**Confidence:** 3

**Summary:**

In this work, the authors use a new theoretical framework for analyzing learning under label noise in multi-class classification.
The proposed framework is based on relative signal strength (RSS), which measures noisiness data points in the training sets.
Based on RSS, the authors propose new upper and lower bounds on excess risk and identify when the classifier learned from label noise is consistent. Based on the theoretical results, a new simple learning framework, called Noise Ignorant Empirical Risk Minimisation (NI-ERM), is proposed, which basically performs standard ERM learning on nosy data. To practically apply NI-ERM, the authors proposed a simple framework of learning linear classifiers on top of a feature extractor trained in an unsupervised/semi-supervised way. The effectiveness of this approach is validated in a few experiments where the method was tested on popular benchmarks under different noise distributions.

**Strengths:**

- The paper is sound and, despite being very theoretical, is easy to read, as authors step by step explain the thinking process.
- Newly obtained bounds are indeed simple in form.
- The effectiveness of the new two-step NI-ERM is confirmed by a few empirical empirical experiments.

**Weaknesses:**

- It seems to me, that the analysis conducted under the Relative Signal Strength framework does not provide a lot of new surprising insights, the conclusions more or less confirm findings from previous works.
- Training feature extractors in an unsupervised/semi-supervised way might be more difficult in the case of some more specialized applications than, for example, general image classification used in the experiments.
- The considered analysis and proposed approach is limited to classifier accuracy, while in many applications other task losses/utilities are often considered.

**Questions:**

Not really, I will be happy to read the authors comments on my points from the weakness section.

**Limitations:**

Limitations were discussed. I see no negative social impact of this work.

---

> ### Author Rebuttal · Authors · 2024-08-05
>
> Thank you for taking the time to review our paper.
> We are especially glad that you find our paper easy to read. Below we would like to address some concerns.
>
> > It seems to me, that the analysis conducted under the Relative Signal Strength framework does not provide a lot of new surprising insights, the conclusions more or less confirm findings from previous works.
>
> Would you be willing to provide specific references to prior work to help us frame our response? Thanks.
>
> We agree that the empirical idea of ignoring the noise is not original (e.g., [1]).
> Theoretically, however, to our knowledge no prior published articles have proved the optimality of the noise-ignorance approach. We are also unaware of any lower bound analysis on label noise that treat a setting as general as ours.
>
>
>
> > Training feature extractors in an unsupervised/semi-supervised way might be more difficult in the case of some more specialized applications than, for example, general image classification used in the experiments.
>
> Fortunately, foundation models are being developed for a rapidly expanding list of application domains, including audio, video, graphs, text and tabular data ([3], Section 4). Generalizable strategies for self-supervised learning are also advancing at a rapid pace, e.g., self-distillation, masked text/image modeling ([3], Section 2).
>
> > The considered analysis and proposed approach is limited to classifier accuracy, while in many applications other task losses/utilities are often considered.
>
> Our results can be extended to the balanced error, and we expect that they also extend naturally to cost-sensitive 0/1 loss. Extensions of our work to other performance measures would be an important research question.
>
> Reference:
>
> [1] Aritra Ghosh and Andrew Lan. Contrastive learning improves model robustness under label noise. In Proceedings of the IEEE/CVF Conference on Computer Vision and Pattern Recognition, 2021.
>
> [2] Ruixuan Xiao, et al. Promix: Combating label noise via maximizing clean sample utility. In Proceedings of the Thirty-Second International Joint Conference on Artificial Intelligence, 2023.
>
> [3] Jonas Geiping, Quentin Garrido, Pierre Fernandez, Amir Bar, Hamed Pirsiavash, Yann LeCun, and Micah Goldblum. A cookbook of self-supervised learning. arXiv preprint arXiv:2304.12210,2023.

---

### Official Review · Reviewer_rXjd · 2024-07-23

**Soundness:** 3
**Presentation:** 3
**Contribution:** 2
**Rating:** 4
**Confidence:** 4

**Summary:**

This paper investigates multiclass classification under label noise, specifically instance-dependent label noise in which the noise can depend on the features as well (a.k.a. local noise). A minimax result is derived that lower-bounds the misclassification probability of a classifier. The paper finds a good empirical practice to be the procedure of training a linear classifier on some separately-learned features.

**Strengths:**

The paper's analysis is correct, and the results relative to the CIFAR-N leaderboard are impressive. (I wish I was able to see the leaderboard, however; several browsers failed to load more than a blank screen.) The message of the paper -- in the presence of unknown noise, learning final-layer classifiers is more robust than learning a classifier from scratch -- is an old one, known empirically to be worth using. The paper's message is clear, and it is overall clearly written.

**Weaknesses:**

The problem addressed in this paper is very common, and people use many heuristic practical strategies it. The paper states that there are theoretical contributions leading to practical consequences. These practical consequences (recommending ignoring the noise) are not new, and the theory lends no understanding on when this might succeed and fail.

The practical consequences are to encourage linear probing (learning a linear classifier on top of a frozen feature layer) on features learned separately from possibly different data. This is possibly the most commonly used idea in practice, including in many situations where it's assumed that there is no noise by default, even when there is. This is not benchmarked thoroughly on different noise distributions. Many other papers in similar areas use Dirichlet distributions to precisely simulate different non-uniform noise, which would be a more thorough synthetic benchmark (e.g. Garg et al. "A Unified View of Label Shift Estimation"). On the real-data side, I would like to see different fine-tuning strategies also compared.

The theory is quite brittle in a couple of ways. It only takes into account the gap to the top class probability, making some of the definitions like A_0 brittle as well; for many classes of noise distributions A_0 could be quite small to empty. The minimax lower bound construction again uses uniform label noise, meaning that it does not adapt to the actual structure of the instance-dependent label noise; the results showing NI-ERM matching this bound are therefore fairly weak.

Presentation-wise, the paper introduces many new terms without apparent reason; these are used correctly, but their combined effect is not to simplify and may be misleading as to the results. An example is "immunity" - it is easy to construct class-conditional noise distributions which would break the minimax results, and they tend to be non-uniform in interesting ways.

**Questions:**

- Why not simulate with instance-dependent label noise of some kind (e.g. simulate cluster-dependent label noise) instead of uniform label noise? Uniform label noise is easy to combat - simply being oblivious works.

- The main message of the paper is that linear probing deals with label noise better than full training / fine-tuning of the network. Are the authors aware of the two-step strategy of linear probing then fine-tuning the whole network, as popularized recently as a general strategy for encouraging robustness to distribution shift? How does this do? (ref. Kumar et al. '22, "Fine-Tuning can Distort Pretrained Features and Underperform Out-of-Distribution")

- How about estimating label shift and benchmarking similar methods, which all implicitly rely on the adjustments to the calibration of a classifier? The Garg et al. reference given above, and methods therein, are SOTA here, and I would think that if there is significant class-conditional noise in \tilde{Y}, then estimating this would be key and perform better than NI-ERM.

**Limitations:**

Yes

---

> ### Author Rebuttal · Authors · 2024-08-05
>
> Thank you for taking the time to review our paper. We appreciate your feedback and constructive criticism. In response to your comments, we would like to address your concerns regarding weakness of the theory and the experiments. We believe we can clarify many points.
> ## "Brittleness" of theory
> > The theory is quite brittle in a couple of ways ... The minimax lower bound construction again uses **uniform** label noise, meaning that it does not adapt to the actual structure of the instance-dependent label noise
>
> **No, we did not use uniform label noise in the proof.** See the construction of $\eta(x), \widetilde{\eta}(x)$ in line 437 Eqn. (2) - (4). To be specific, the label noise in that construction puts all probability mass into one specific class, while uniform label noise would spread probability mass into all classes instead.
>
> The reviewer may be alluding to the fact that we "let $J \sim Uniform (1,2, ..., K)$" in line 454, or "let $B \sim Uniform (1,2 )^{V-1}$" in line 468. This is a technique called the "probabilistic method" that is commonly used in the minimax lower bound proofs (see, e.g., [7] Section 14).
>
> We re-iterate that our lower bound proof did not use uniform label noise. The bound applies exactly to **instance-dependent label noise**.
>
> > It only takes into account the gap to the top class probability, making some of the definitions like $A_0$ brittle as well; for many classes of noise distributions $A_0$ could be quite small to empty.
>
>
> If $A_0$ is empty, then our theory still returns a meaningful result. In this case, the noisy and clean Bayes classifiers disagree almost everywhere, and therefore $\epsilon = 1$, leading to a (big) lower bound of $(1-1/K)$ on the clean excess risk. So we are not sure why you say our theory is brittle here, and we welcome clarification.
>
> "the gap to the top class probability" is fundamental in learning theory, see for example the standard assumptions of Massart and Tsybakov [1-3].
>
> ## Practical performance
>
> > The results relative to the CIFAR-N leaderboard are impressive. I wish I was able to see the leaderboard, however; several browsers failed to load more than a blank screen.
>
> We have contacted the research team who maintains the CIFAR-N leaderboard. They have fixed it, welcome to try it again <http://noisylabels.com/>.
> Otherwise, wayback machine records its status on May 23rd, see <https://web.archive.org/web/20240523101740/http://noisylabels.com/>.
>
> > These practical consequences (recommending ignoring the noise) are not new, and the theory lends no understanding on when this might succeed and fail.
>
> We acknowledge that the practical idea of ignoring label noise is not new [4], but the full power of this approach has not been previously recognized. For example, prior work that has suggested ignoring the label noise usually augments this approach with additional heuristics such as fine-tuning with early stopping [5]. We will add a paragraph about previous works that practically suggest ignoring the noise.
>
> Furthermore, our theory precisely describes when the method succeeds: It is nearly minimax optimal under the setting of Thm 2, and consistent under the settings discussed in Section 5.
>
> > How about estimating label shift ...  The Garg et al. reference given above, and methods therein, are SOTA here, and I would think that if there is significant class-conditional noise in $\tilde{Y}$, then estimating this would be key and perform better than NI-ERM.
>
> **Label shift is a different problem (compared with label noise).** In label shift, the X-marginal distribution changes (from source to target) whereas with label noise, it does not. Furthermore, label shift methods assume access to an unlabeled test dataset, which is also not present in the label noise setting. Finally, label shift methods estimate the class prior for the test data (which is a $K$ dimensional vector), not class-conditional noise parameters in a class-conditional noise model (which is a $K \times K$ matrix).
>
> Or, are you asking if "label shift-ignorant ERM" works in a label shift problem? It sounds like an interesting research question and worth exploring. Again we welcome clarification.
>
> > Are the authors aware of the two-step strategy of linear probing then fine-tuning the whole network... How does this do?
>
> Yes, we are. We have ran additional experiments: **LP-FT is generally beneficial when there is no label noise, but becomes harmful when noise rate increases**, see Table 1 in the pdf attached to the global rebuttal.
>
> > Why not simulate with instance-dependent label noise of some kind (e.g. simulate cluster-dependent label noise) instead of uniform label noise?
>
> We have ran additional experiments: we simulated instance-dependent label noise using a commonly adopted approach in [8]. Our NI-ERM performs better than the approach proposed in [8]. See Table 3 in the pdf attached to the global rebuttal for detail.
>
> Reference:
>
> [1] Alexander B Tsybakov. Optimal aggregation of classifiers in statistical learning. The Annals of Statistics, 2004.
>
> [2] Pascal Massart and Élodie Nédélec. Risk bounds for statistical learning. The Annals of Statistics, 2006.
>
> [3] Vu Dinh, et al. Learning from non-iid data: Fast rates for the one-vs-all multiclass plug-in classifiers, TAMC 2015.
>
> [4] Aritra Ghosh and Andrew Lan. Contrastive learning improves model robustness under label noise. CVPR Workshop, 2021.
>
> [5] Yihao Xue, et al. Investigating why contrastive learning benefits robustness against label noise. ICML, 2022.
>
> [6] Ruixuan Xiao, et al. Promix: Combating label noise via maximizing clean sample utility. IJCAI, 2023.
>
> [7] Luc Devroye, László Györfi, and Gabor Lugosi. A Probabilistic Theory of Pattern Recognition, 1996.
>
> [8] Xiaobo Xia, et al. Part-dependent label noise: Towards instance-dependent
> label noise. Advances in Neural Information Processing Systems, 2020.

---

> ### Comment · Reviewer_rXjd · 2024-08-12
> **Good rebuttal, concerns remain**
>
> Thanks to the authors for their detailed and precise comments. As I write below, I agree with the authors on many points (and have changed my review accordingly), but would leave much of my initial assessment intact.
>
> In my initial review, I mis-spoke on a couple of points, which I will clarify. The lower bound construction does indeed use instance-dependent label noise on a discrete space, not uniform label noise. Label shift is indeed a different problem, and I do not mean to suggest this is the same setting; but the simulations/benchmarking of the papers in that area comes far closer to practical relevance here. Thanks to the authors for their experiments in the rebuttal, including on LP-FT.
>
> The NI-ERM method itself is oblivious of the existence of any noise, and there is no new algorithm or loss function. Yet it is standard practice because it seems to sometimes work. So the paper's contribution is looking to justify where the standard procedure does and doesn't work - in which label noise situations would it be fine to use, and where should one look to do something else - where is ignorance bliss?
>
> This is why the brittleness of the theory matters in this context. I am very familiar with the standard label noise literature (Tsybakov, Massart, et al.) and the authors are correct about the theory here also being in that spirit. However, that theory is most robust/successful (and originally developed) for binary classification, in which the problem reduces to one where the authors' RSS is a good pointwise signal-to-noise measure. As K > 2, the min over the K classes in the RSS starts to eliminate more and more information, and lower the RSS even when learning would appear to be intuitively possible. For e.g. Imagenet (K=1000) or many similar-scale problems, the classes themselves show significant structure, with some being easily confusable for each other, and others less so; the sets A_{\kappa} could be small or even null, in my practical opinion. Clearly this will not nontrivially happen for K = 2, and tends to happen less for low K.
>
> Ideally a reasonable \kappa (to get a reasonable \epsilon) should scale nicely with K, which would make the upper bounds quite relevant. I don't have good intuition about whether this applies for noise distributions in the wild. (It would be great to see if this is correct by computing A_{\kappa} for a real dataset, which can be done approximately with sampling. If A_{kappa} is nontrivial for reasonable \kappa; the inverse dependence of the bounds on \kappa means that we do need some statistically nontrivial gap).
>
> This of course affects the model class over which the minimax result is proved. The result is strong (like any minimax result is) *over its posterior drift model class* - but \Pi may be nearly empty. The lower bound construction is certainly correct, but its tightness, and the logic of the proof, again rely on the model class \Pi.
>
> All this means I am not really able to answer the question, as a practical matter, of when ignorance is bliss. The practical relevance of the theory here appears limited for higher K (though it could be very relevant for low K). There are of course many theories of learning under label noise that extend beyond the binary setting, but those often lead to a new algorithm, loss function, etc. Since this is not true here, the "practically interpretable" aspects of the theory are important in assessing the contribution.

---

> ### Author Response · Authors · 2024-08-13
>
> We thank reviewer for the detailed response.
>
> ## Contribution of the paper
>
> > ... so the paper's contribution is looking to justify where the [Noise ignorant] procedure does and does not work.
>
> Actually, we feel that this is not our main focus (although the exploration of section 5 is concerned with this, e.g., Thm 4).
>
> The main theoretical focus of the paper is **what is the fundamental limit (e.g., minimax risk) in learning with label noise**. In other words, if we **do not assume any further structure on the data**, what can we hope to get, and what works for this. It is in this regard that Section 4 (and the preceding definitions of Section 3) are strong: they identify that the minimax risk is captured directly by the RSS, and that the RSS is the 'right' object to think about when considering label noise.
>
> Note that the RSS is a novel way of characterising the noisy labels problem. The framework not only admits tight bounds, suggesting that at least in a minimax sense RSS is the natural object to study, it also is flexible enough to capture structure (of label noise), one example is demonstrated through Thm 3.
>
> ## What if number of classes $K$ is big?
>
> > As $K > 2$, the min over the K classes in the RSS starts to eliminate more and more information, and lower the RSS even when learning would appear to be intuitively possible. For e.g. Imagenet ($K=1000$) or many similar-scale problems, the classes themselves show significant structure, with some being easily confusable for each other, and others less so... Ideally a reasonable $\kappa$ (to get a reasonable $\epsilon$) should scale nicely with K
>
> When $K$ becomes large, even the standard learning scenario becomes provably hard (e.g., literature on extreme classification).
> We imagine the same also holds for label noise setup.
>
> As for a solution for this, just as you have suggested, is to explore 'structure', formally speaking, assume **sparsity**: both $\eta, \widetilde{\eta}$ are $C$-sparse, which corresponds to the scenario you described as "...some (classes) being easily confusable for each other, and others less so".
>
> Now, let us examine the concept of RSS again:
> $$
> M(x; \eta, \widetilde{\eta}) = \min_{j \in \\{1, 2, \dots, K\\} } \frac{ \max_i [\widetilde{\eta}(x)]_i - [\widetilde{\eta}(x)]_j }{ \max_i [\eta(x)]_i - [\eta(x)]_j },
> $$
>
> For the "unrelated classes" $j$, where both $[\widetilde{\eta}(x)]_j$ and $[\eta(x)]_j$ are zero,
> $$
> \frac{ \max_i [\widetilde{\eta}(x)]_i - [\widetilde{\eta}(x)]_j }{ \max_i [\eta(x)]_i - [\eta(x)]_j } = \frac{ \max_i [\widetilde{\eta}(x)]_i }{ \max_i [\eta(x)]_i },
> $$
> thus it will not deflate the RSS.
>
> Therefore, the minimum of $j$ over $\\{1, 2, \dots K\\}$ reduces to the set containing non-zero indices of $\eta$ and $\widetilde{\eta}$, which contains at most $2C$ elements (from the $C$-sparse assumption above).
>
> To conclude, with an additional sparisty assumption which takes the noise structure into account, our notion of RSS "scales well with $K$".
>
>
> ## $A_0$ being null?
>
> > For e.g. Imagenet (K=1000) ... with some (classes) being easily confusable for each other, and others less so; the sets $A_{\kappa}$ could be small or even null, in my practical opinion.
>
> In order to claim "$A_{\kappa}$ could be small or even null", several things need to be taken into account:
> - value of $\kappa$
> - overall hardness of the classification task: $\eta(x)$
> - quality of labeller: $\widetilde{\eta}(x)$
>
> For an un-trained labeler (e.g., common people like you and us), images of some classes are "easily confusable", thus the resulting $A_0$ can be quite small.
>
> For the "ImageNet labeler(s)" who have gone through detailed instruction and careful quality control, the (final) label quality is much better, thus $A_0$ shall be quite big.
>
> For a fixed unlabeled image dataset, and a fixed value of $\kappa$, how big $A_{\kappa}$ depends on (human) label quality, which shall be analyzed case-by-case.
>
> Towards this end, we thank you for bringing up the task of "calculating the RSS level of real-world noisy dataset". We could start from the CIFAR-N dataset. This is a big project, beyond the scope of the current paper, but would benefit from the framework constructed in this paper.
>
> ---
>
> To conclude, the main point of the paper is not to "justify when NI-ERM does and does not work", but to develop a theoretical understanding of the general label noise problem (through the new concept of RSS). We thank you for bringing up the question on how the theoretical concept can be used to explain "label noise in the wild", we believe it is important and worth studying. Thank you again.

---

### Author Rebuttal · Authors · 2024-08-06

We thank all reviewers for your time and effort. Our paper will be substantially better as a result of your comments and questions. Here we provide responses to questions that seem most likely to be of interest to all reviewers.

## Theory

> KL divergence vs. Relative Signal Strength (RSS)

We thank Reviewer G7Jo for bringing up the question on the relation of KL divergence and our new concept RSS. We will include a new paragraph and a new example in our next version to illustrate this. This shall make the concept of RSS more intuitive.

The short answer is: KL divergence considers the similarity between two (whole) distributions, while the task of classification only focuses on predicting the $\arg \max$, making RSS the correct measure. See our response to G7Jo for more detail.

> uses "uniform label noise" in minimax lower bound proof?

No, we did not use uniform label noise in the proof. Our lower bound holds for general instance-dependent label noise. For details, see our response to Reviewer rXjd.

## Experiments

> include comparison to more methods, apply SOTA methods to the same high quality features, linear probing - then fine tuning (LP-FT), ...

These are reasonable requests. We have run additional experiments comparing our NI-ERM approach to previously proposed "noise robust" methods, see **attached pdf** file for details.

In Table 1, we report results for LP-FT (linear probing then fine tuning), and find that it underperforms our approach when noise is present.

In Table 2, we compare against two recently published robust training methods by giving those methods access to the same high-quality DINOv2 features that we used. Again we find that NI-ERM is superior. We also reran our own method using some robust losses (mean absolute error, sigmoid) and find that these are comparable to cross-entropy (which we used for the submission).

In Table 3, we ran experiments using a synthetically generated, non-uniform instance-dependent label noise. Again, NI-ERM is superior.

We will add of these results to the next version of the paper.

We thank all reviewers again.

---

### Author Response · Authors · 2024-08-11

Dear area chair,

Up to now we have not received any comments from reviewers.

Could you please ask the reviewers to engage with us?

Thanks.

---

> ### Author Response · Authors · 2024-08-14
>
> We thank all reviewers for the feedback.
> The discussion is very helpful.

---

### Decision · Program_Chairs · 2024-09-25

**Decision:**

Accept (poster)

**Comment:**

The paper presents a novel theoretical framework for learning with multi-class, instance-dependent label noise. While the reviewers were somewhat divided, the overall decision is to accept the paper.